

# Review article: Melt-Affected Ice Cores for (Sub-)Polar Research in a Warming World

Dorothea Elisabeth Moser[1,2], Elizabeth R. Thomas[1], Christoph Nehrbass-Ahles[2,3], Anja Eichler[4,5], Eric Wolff[2]

[1]Ice Dynamics and Palaeoclimate, British Antarctic Survey, Cambridge, CB3 0ET, United Kingdom
[2]Department of Earth Sciences, University of Cambridge, Cambridge, CB2 3EQ, United Kingdom
[3]Gas Metrology Group, National Physical Laboratory, Teddington, TW11 0LW, United Kingdom
[4]Laboratory of Environmental Chemistry, Paul Scherrer Institute, CH-5232 Villigen PSI, Switzerland
[5]Oeschger Centre for Climate Change Research, University of Bern, CH-3012 Bern, Switzerland

*Correspondence to*: Dorothea Elisabeth Moser (moser@bas.ac.uk)

**Abstract.** Melting polar and alpine ice sheets in response to global warming pose ecological and societal risks but will also hamper our ability to reconstruct past climate and atmospheric composition across the globe. Since coastal low-elevation ice caps are crucial environmental archives but changing rapidly, the (sub-)polar research community is increasingly faced with melt-affected ice cores common in alpine settings. Here, we review the characteristics and effects of near-surface melting on ice-core records focussing on a (sub-)polar readership and make recommendations for melt-prone study regions. This review covers (1) melt layer formation; (2) identification and quantification of melt; (3) structural characteristics of melt features; effects of melting on (4) records of chemical impurities, i.e. major ions, trace elements, black carbon, and organic species, (5) stable water isotopic signatures, and (6) gas record; (7) applications of melt layers as environmental proxies.

Melting occurs during positive surface energy balance events, which are shaped by global to local meteorological forcing, regional orography, glacier surface conditions and subsurface characteristics. Meltwater flow ranges from homogeneous wetting to spatially heterogeneous preferential flow paths and is determined by temperature, thermal conductivity and stratigraphy of the snowpack. Melt layers and lenses are the most common consequent features in ice cores and are usually recorded manually or using line-scanning. Chemical ice-core proxy records of water-soluble species are generally less preserved than insoluble particles such as black carbon or mineral dust due to their strong elution behaviour during percolation. However, a high solubility in ice as observed for ions like $F^-$, $Cl^-$, $NH_4^+$, or ultra-trace elements can counteract the high mobility of these species due to a burial in the ice interior. Stable water isotope records like $\delta^{18}O$ are often preserved but appear smoothed if significant amounts of meltwater were involved. Melt-affected ice cores are further faced with questions about the permeability of the firn column for gas movement, and gas concentrations can be increased through dissolution and in-situ production. Noble gas ratios can be useful tools to identify melt-affected profile sections in deep ice. Despite challenges for ice-core climate reconstruction based on chemical records, melt layers are a proxy of warm temperatures above freezing, which is most sensitive in the dry snow and percolation zone.



Bringing together insights from snow physics, firn hydrology and ice-core proxy research, we aim to foster a more comprehensive understanding of ice cores as climate and environmental archives, provide a reference on how to approach melt-affected records, and raise awareness of the limitations and potential of melt layers in ice cores.

## 1 Introduction – The importance of studying melting and affected ice cores

Polar ice cores are widely recognised for their ability to archive past climates on local to global and sub-annual to multi-millennial scales (EPICA Community Members, 2004; NEEM community members, 2013; WAIS Divide Project Members, 2013). Beyond the polar regions, ice cores from alpine glaciers, ice fields or lower elevation coastal domes and islands (Fritzsche et al., 2005; Porter et al., 2021; Thomas et al., 2021; Winski et al., 2018) provide an opportunity to reconstruct more local and regional climatic change. Rising global temperatures (IPCC, 2021) and accelerated melting of the most vulnerable parts of the cryosphere (IPCC, 2019) cause glaciers across the globe to shrink, thereby hampering our ability to reconstruct past climate from ice cores across the globe.

On the snow surface of a glacier, melting and refreezing leaves an imprint on the snowpack, i.e. typically cm-thick, bubble-free melt lenses and layers with fuzzy edges (Figure 1). Wind abrasion, deposition hiatus and other factors, which are still under discussion, can also lead to distinct, mm-thick, high-density bubble-free layers like wind crusts and glaze that are structurally similar to melt layers (Orsi et al., 2015; Fegyveresi et al., 2018; Frezzotti et al., 2002; Zhang et al., in review.; Weinhart et al., 2021), but water is absent during the formation process. Presence and impact of the water phase are what makes melt a major constraint for the selection of ice-core drilling sites. Unlike the original climate signatures preserved under dry conditions, palaeoclimate reconstructions based on ice cores from melt-affected drilling sites require caution and have been hampered by insufficient knowledge about the effects of melting on ice-core proxy records and how to correct for them in the past. In recent decades, alpine ice-core scientists faced with this issue regularly have made significant advances regarding the interpretation of melt-affected ice-core records (Avak et al., 2018). Improvements have also been made on the wider subject of snow melt with interdisciplinary methodical approaches ranging from detection with remote sensing (Alimasi et al., 2020; Bell et al., 2017; Dell et al., 2021) to field-based snow pits (Avak et al., 2019; Moran & Marshall, 2009; Spolaor et al., 2021), firn sintering and elution experiments (Cragin et al., 1996; Trachsel et al., 2019), modelling (Zhongyang et al., 2021), and ice-core studies (Avak et al., 2018; Orsi et al., 2015; Vega et al., 2016). Yet among polar ice-core scientists, still only limited agreement exists about the informational value of melt-affected ice cores: On the one hand, chemical records of refrozen melt layers are often considered lost or altered, and likely to lead to misinterpretations of proxy records (Koerner, 1997). On the other hand, these structural features have been used as proxies for warm summer temperatures in multiple studies from the Arctic and Antarctic (Abram et al., 2013; Das & Alley, 2008; Koerner & Fisher, 1990).

In the past, studies about melt in ice cores were focussed on alpine settings (Avak et al., 2018; Eichler et al., 2001; Kang et al., 2008; Yuanqing et al., 2001). Such mountain glaciers around the Earth are vanishing (IPCC, 2019), because a rise in local equilibrium lines driven by global heating exposes most of their surfaces to melt as a cause of ablation. At the extreme end,





the fear that melt may make ice cores from some sites unusable has motivated the Ice Memory Initiative (Ginot et al., 2021).

Among (sub-)polar ice-core researchers, increasing temperatures have sparked interest in drivers of snow melt and firn hydrology in the Arctic and Antarctic, where the firn is generally cold even though melting may occur. For example, in Greenland, the extensive melt event in 2012 (Nghiem et al., 2012) was followed up by record-melt in 2019 (Tedesco & Fettweis, 2020) and rain-driven melt reached Greenland Summit in August 2021 (Box et al., 2022). The increase in Greenland ice sheet melting (IPCC, 2019) and rainfall (Niwano et al., 2021) is expected to continue in a warming climate. Even in

Antarctica, recent melt events (Nicolas et al., 2017) have raised concerns about a reduction of firn refreezing capacity, potential hydro-fracturing and ice shelf instability (Bell et al., 2017; Gilbert & Kittel, 2021; Munneke et al., 2014; Wille et al., 2019). Surface melt events are rare in Antarctica and mostly restricted to the Antarctic Peninsula (Van Wessem et al., 2016), coastal (Nicolas et al., 2017), low-elevation sites (Thomas et al., 2021) and ice shelves (Banwell et al., 2021; Zou et al., 2021). However, those regions are most sensitive to climate change, and assessing the frequency and characteristics of Antarctic and

sub-Antarctic melt events is crucial to constrain the (surface) mass balance of Antarctica and future sea level rise at large. In addition, the amplified warming of the polar regions is expected to continue and even accelerate depending on the greenhouse gas concentration trajectories in the next few decades (Trusel et al., 2015). Though total surface melting seems to have decreased slightly in the Antarctic Peninsula in the first two decades of the 21st century, winter melt events are becoming more frequent (Zheng et al., 2020), and the global temperature rise leading to melt will affect a growing number of alpine to polar

ice-core drilling locations, more frequently, with larger intensity, and at higher elevation further inland (Clem et al., 2020; Gilbert & Kittel, 2021; Nicolas et al., 2017; Winski et al., 2018).

Bridging the gap between alpine expertise on how to handle melt in ice cores and less melt-acquainted polar ice-core scientists is urgently needed, because understanding the drivers of recent climate changes, in particular increased melt and melt-driven changes in Antarctic mass balance is paramount. With increasing interest in the role of sea ice (Thomas et al., 2019),

atmospheric circulation (Wille et al., 2019), and tropical teleconnections (Nicolas et al., 2017), the research field is increasingly looking to the fringes of the Antarctic ice sheet. The demand for (sub-)polar ice-core profiles from low elevations, coastal and island regions means that effects of melt on ice-climate proxy records cannot always be avoided. The polar ice-core community urgently needs to prepare for ice-core science under warming conditions.

It is the aim of this review to bring together insights from snow physics, firn hydrology and ice-core proxy research in order

to present the status quo on the influence of melt on ice proxy preservation, and subsequently make suggestions for future projects in melt-prone study regions. We provide a detailed literature review regarding (1) external drivers of melt events; (2) physics of melt layer formation and behaviour during snow metamorphism; (3) identification and quantification of melt; (4) structural characteristics of melt features; effects of melting on (5) records of chemical impurities, i.e. major ions, trace elements, black carbon (BC), and organic species, (6) stable water isotopic signatures, and (7) gas record; (8) applications of

melt layers as environmental proxies. We, thereby, focus on those aspects of near-surface melting, which are important for ice-core research. Englacial or basal melt phenomena as well as glacier-wide melt assessment based on remote sensing are



beyond the scope of this study. With this study, we aim to learn from alpine ice-core scientists and contribute to a more comprehensive understanding among polar researchers of the limitations and potential of melt features in ice cores.

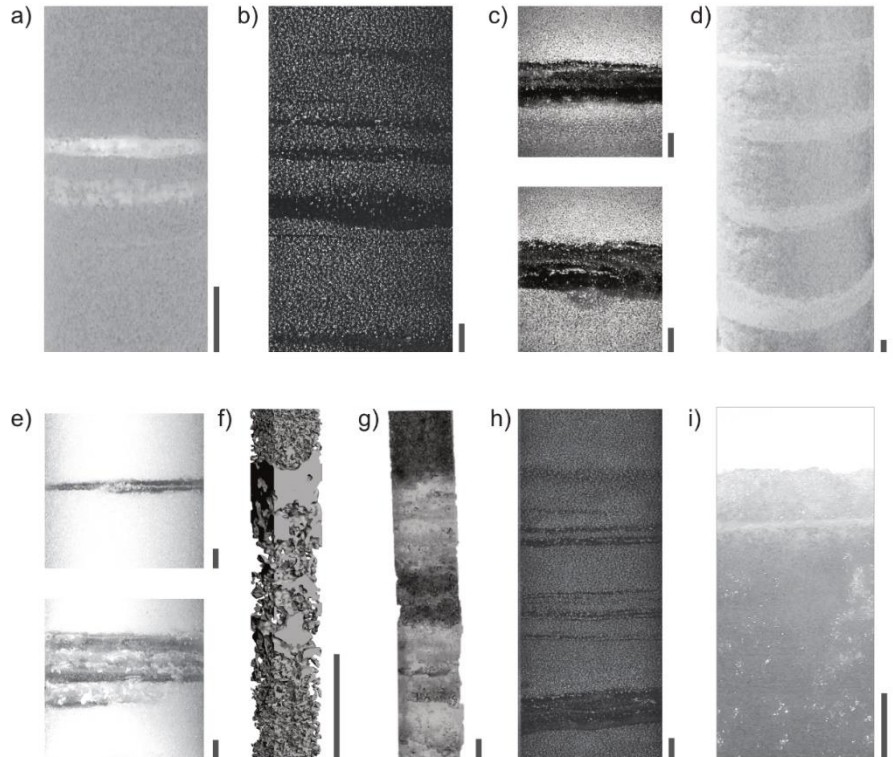

**Figure 1: Examples of melt layers reported in snow and ice profiles from around the world using various techniques; a) two melt layers, 4–6 mm thick, from the NEEM Greenland ice core at ~44.3 m depth (digital photo, credit Kaitlin Keegan, Orsi et al., 2015); b) sequence of melt layers ranging 3–13 mm at EastGRIP, Greenland ice core at ~138.04 m depth, (line scan image, Westhoff et al., 2022, CC-BY 4.0); c) two distinct melt layers ~15-20 mm thick at RECAP, Renland Ice Cap, at 78.1–79.2 m depth (line scan image, Taranczewski et al., 2019); d) four ~20–35 mm melt layers at 0.3–0.5 m depth in an ice core from the Agassiz Ice Cap, Canada (digital photo, credit James Zheng, Fisher et al., 2012); e) varying appearance of 3–7 mm and 25–41 mm melt layers in an ice core from Mount Hunter, Alaska (digital photo, Winski et al., 2018); f) ~15–20 mm melt at ~0.04 m depth in surface snowpack from Davos, Swiss Alps (3D X-ray tomography scan, Heggli et al., 2011); g) segment at ~3 m depth from the Young Island ice core with melt >45 mm thick (digital photo, Moser et al., 2021, CC-BY 4.0); h) melt sequence between 133.82–1.92 m depth in ice core from James Ross Island, Antarctic Peninsula (line scan image, Abram et al., 2013); i) a 1-mm melt layer in the surface snow at 0.008 m depth at Siple Dome, Antarctica (digital photo, Das and Alley, 2005); the grey scale bar at the bottom right of each panel equals 1 cm; the top of each snow/firn/ice section is at the top of each panel.**

## 2 From melting to melt-affected ice cores

### 2.1. Drivers of melt

Generally, melting takes place whenever sufficient energy is available to cause a phase change from ice to water. The components of this surface energy balance (SEB) contributing to this surplus can be expressed as shown in the following Equation (Eq. 1):





$$M = SW_\downarrow - SW_\uparrow + LW_\downarrow - LW_\uparrow + H + L + R + G \qquad \text{(Eq. 1)}$$

where M denotes the energy available for heating ice at sub-zero temperatures or melting temperate ice, SW refers to incoming (↓) and outgoing (↑) shortwave radiation; LW is longwave radiation; turbulent fluxes comprise sensible heat flux (H) and latent heat flux of water vapour (L); R is the sensible energy supplied by rain, and G is the ground heat flux. Numerous studies have assessed the surface energy balance and how to model the underlying processes (Hock, 2005). Here, we focus on the environmental context of the SEB, because each component of the SEB is driven by global-to-local meteorological forcing (Kinnard et al., 2008) and its interactions with the surface and sub-surface properties of the glacier (Fig. 2).



**Figure 2: Schematic overview of the diverse, external drivers and constraints of near-surface snow melt discussed in Sect. 2.1., including global to local meteorological forcing, regional orography, surface conditions, and subsurface characteristics.**

Positive air temperatures are the primary trigger of melt and therefore a well-established, integrated proxy for melt intensity (Abram et al., 2013; Hock, 2003). The relationship between temperature and melting of the snowpack is non-linear (Bell et al., 2018). For example, a 1°C temperature increase of snowpack's temperature from -21°C to -20°C does not have the same



physicochemical implications for ice-core records as the rise from 0°C to +1°C, the latter being connected with a phase change from ice to water.

Warm air can be linked to various components of the SEB. Melt rates of up to 4 mm per day have been recorded on the Eastern Antarctic Peninsula, attributed to warm air advection in combination with foehn wind effects (Van Den Broeke, 2005). Warm air advection is shaped by synoptic-scale atmospheric circulation patterns like high-pressure blockages, which facilitate moist-

warm air intrusions (e.g. West Antarctic Ice Sheet, WAIS, Scott et al., 2019). Similar importance of atmospheric blocking has been observed for Greenland melt (Hanna et al., 2013, 2014; McLeod & Mote, 2016). Linear-shaped air intrusions are often referred to as 'atmospheric rivers' and correlate highly with melt production, especially during winter. Atmospheric rivers have been considered responsible for the observed extensive melt in West Antarctica in 2016 (Wille et al., 2019) as well as in Greenland (Neff, 2018). The atmospheric forcing of melt events also includes near-surface winds (Cullather & Nowicki, 2018).

Foehn events are characterized by clear-sky shortwave radiation together with advection of warm, dry air and occur episodically by gravity winds on the lee slopes of mountain ranges like the Antarctic Peninsula (Elvidge & Renfrew, 2016). These foehn winds are coupled to position and strength of the circumpolar westerly wind belt (Datta et al., 2019) and can lead to spatially inhomogeneous melt events (Bell et al., 2018). The break-up of Larsen B ice shelf was influenced by foehn-driven melt events, which peak in frequency during autumn (Cape et al., 2015; Datta et al., 2019) but are not limited to this season

and become more frequent as the polar vortex contracts during the current positive phase of the Southern Annular Mode (SAM, Marshall et al., 2006). Melting can also be caused through the 'blanketing effect' of a cloud cover (Cullather & Nowicki, 2018; Hahn et al., 2020; Scott et al., 2019), which increases the down-welling longwave radiation balance. In this regard, altitude and phase-type of the clouds prevailing at the site play a key role for the net radiation effect. Low-altitude, mixed-phase clouds warm the glacier surface, and vice versa.

Such meteorological constellations in the Antarctic are dynamically tied to the climate of the tropics (Winski et al., 2018). When examining melt in ice cores, even such long-distance teleconnections should be taken into account. For example, the strength and position of the Amundsen Sea Low (ASL) is coupled to tropical climate through the El Niño/Southern Oscillation (ENSO) teleconnection, and summer melt in West Antarctica has been linked to strong El Niño (Nicolas et al., 2017; Wille et al., 2019).

Besides the meteorological and climatological drivers of melt events, the regional-to-local shape of an ice body determine its melt response. As the lateral and vertical extent of an ice sheet affects the trajectories of air-masses circulating around it, ice sheet geometry is to be considered a factor for melt occurrence (Das & Alley, 2008). Melt conditions can also vary across short distances, because the regional-scale meteorological drivers described above interact with local site characteristics (Fig. 2). Glacier orography including slope, aspect, and elevation plays a significant role for the local SEB (Arnold et al., 2006;

Hahn et al., 2020; Olson & Rupper, 2019). For example, the warming of katabatic winds during its downflow has been shown to enhance melt production especially at the grounding line of an East Antarctic glacier during summer, facilitated by positive wind-albedo and melt-albedo feedbacks through the exposure of blue ice and bedrock (Bell et al., 2018; Lenaerts et al., 2017).



Melt is further driven by site-specific surface snow characteristics (Fig. 2). Snow exposed at the surface undergoes processes like wind-driven redistribution, temperature-gradient and isothermal metamorphism, which alter the size and shape of the
deposit and in turn surface albedo. While fresh snow exhibits albedo values of ~0.8, aged snow shows lower values around 0.4 (Dirmhirn & Eaton, 1975) and dirty snow even lower values <0.2 (Conway et al., 1996; Warren, 2019). Thereby, albedo is coupled to melt by positive feedback (Thackeray & Fletcher, 2016), i.e. a lowering of the surface albedo reinforces the tendency for melting and leads to a further albedo decrease. For example, forest fires produce significant amounts of BC and reduce the surface albedo even at distant sites like interior Greenland, a coupling that is expected to increase with climate change (Keegan
et al., 2014). Also, debris cover and cryoconite (Fountain et al., 2004), dark-coloured dust containing soot and microorganisms, can lower the albedo and enhance local melt production (Mattson et al., 1993). A debris cover can also act as an insulating active layer, if it is sufficiently thick (Reznichenko et al., 2010; Rowan et al., 2021).

All of this shows that the drivers of a positive SEB and melt events are not uniform. They must be assessed for individual sites and ice-core records within their respective, complex climatic context. The details of the melt process and its sub-surface
progression shall be discussed in Sect. 2.2.

## 2.2. Physics of melt layer formation and dynamics during snow metamorphism

The analysis of melt-affected ice core records is preceded by understanding the physicochemical process of melting ice, which has been investigated extensively since the 19[th] century (Faraday, 1859; Sánchez et al., 2017). Sánchez et al. (2017) recently presented evidence that melting takes place in a 'bilayer-by-bilayer' fashion leading to a nanometre-thick quasi-liquid layer
(QLL) at the ice-air interface (Slater & Michaelides, 2019). As these features already exist within the ice matrix on a grain scale at temperatures below the melting point of the wider snowpack, they are also called 'pre-melting' (Bishop et al., 2009). On a larger scale, the surface snow starts to melt when the energy balance is positive sufficiently long enough to raise at least parts of the ice matrix to 0°C, and it ends when the energy balance below the 0°C isotherm is negative again. This duration threshold is reached within a few hours, as has been recorded in Antarctica by Laska et al. (2016). Therefor modelling of melt
processes must aim for hourly or at least sub-daily resolution. Furthermore, Koh and Jordan (1995) have shown that radiation-forced temperature maxima and melting can occur a few centimetres below the snowpack surface, especially in low-density snow. Since the surface energy conditions vary significantly over short time scales, e.g. during diurnal cycles, 'melt-freeze' features (referred to as melt features hereafter) tend to develop accordingly (Colbeck, 1982).

Melting leads to wet metamorphism, during which liquid water is available and acts as a carrier of thermal energy, which
accelerates post-depositional alteration of the affected snow layers. Wet metamorphism is generally isothermal (Brun, 1989; Wakahama, 1968, 1975), characterized by a rise in grain size and density caused by recrystallization, and increasing liquid water content softens the snow (Colbeck, 1982). This process leads to the key structural characteristics of melt features in ice cores, which is discussed in Sect. 3.1.

Melt generated near the snow surface can penetrate into the underlying snowpack (Westhoff et al., 2022). The progression of
the percolation front depends on local thermodynamic and hydrological conditions and has been modelled and observed to



varying degrees from alpine to polar sites (Graeter et al., 2018; Meyer & Hewitt, 2017). Meltwater infiltration into deeper, older snow makes percolation a secondary process leading to mixed snow compositions and climatic signatures, e.g. of summer melt in winter snow (Moore et al., 2005). Infiltration beyond the current annual layer is a source of uncertainty when interpreting melt layer records, so that using multi-year averages of melt indices has been recommended previously (Graeter

et al., 2018). Furthermore, positive-degree-day (PDD) sums are a common tool to quantify ablation due to temperatures above melting point (Hock, 2003). With PDD values normalized to a single day (°C-day), Das and Alley (2005) have estimated +5.22°C as the temperature for a West Antarctic site, above which it is probable that percolation of a single melt event reaches into accumulation of the previous year. However, this parameter requires site-specific assessment. The rate of melt production further governs the exposure period of snow to melt liquid (Taylor et al., 2001) through the 'saturation, refreezing capacity,

and drainage processes' within the firn (Samimi et al., 2020). This is important when it comes to interpreting chemical records from melt-affected ice cores.

In addition to the energy balance at the snow surface and gravitation in general, speed and type of meltwater flow are determined by characteristics of the subsurface including (1) temperature, (2) thermal conductivity and (3) stratigraphy of the snowpack. (1) Cold subsurface temperatures of the snowpack, the so-called 'cold content', are a strong constraint for snowmelt

production (Jennings et al., 2018). Depending on the cold content, large amounts of energy might be needed to raise the snow temperature to melting point and delay the progression of the melting front at the 0°C isotherm to deeper layers. (2) According to Pfeffer & Humphrey (1998), the balance between melting and refreezing is also driven by thermal conductivity of the ice matrix, which in turn depends on firn density. High densities allow for more freezing, and vice versa. (3) Snow column stratigraphy affects the direction of flow through capillary and hydrological forces. Neighbouring snow deposits can differ

significantly in terms of microstructure, e.g. grain size and porosity, and so do their capillary characteristics and permeability. It is due to this difference in capillary resistance that fine-to-coarse transitions are well known to retard vertical meltwater percolation (Pfeffer & Humphrey, 1996), but differences in capillary characteristics can lead to lateral meltwater suction, too. Hydrological barriers can also be high-density layers like wind crusts (Das & Alley, 2005) or pre-existing melt layers (Samimi et al., 2020), and both are frequent features in alpine to polar settings. Furthermore, progression of melt is not only halted by

fully melt-filled refrozen pores but by closed pore connections (Humphrey et al., 2021). Taken together, snow conditions constraining the flow are expected to lead to sharper contacts between melt layer and adjacent firn.

Based on these factors, which determine the flow path of least resistance, percolation take place in different ways (Hirashima et al., 2019). Homogeneous wetting, also known as matrix flow, tends to occur in temperate snow with less dominant stratigraphic boundaries, and the thawing front progresses laterally uniformly. While at colder, stratigraphically complex sites,

lateral and vertical flow alternate spatially and temporally (Pfeffer & Humphrey, 1996; Williams et al., 2010). Preferential flow takes place where meltwater flow instabilities occur at the wetting front (Hirashima et al., 2014, 2017; Leroux & Pomeroy, 2017; Parlange, 1974). In this case, capillary pressure of advancing vertically exceeds the capillary pressure of expanding laterally. Dye experiments have frequently been used to visualise this stepwise advance of preferential flow paths (PFPs) along the stratigraphy (Avanzi et al., 2016). They also highlight the issue that melt layers don't tend to form directly at the surface





but at 50–100 cm depth (Kaczmarska et al., 2006). Given that the flow conditions are a continuum between these two extremes and vary widely among sites and seasons, different structural characteristics of melt in ice cores must be expected (Sect. 3.1.). Furthermore, ice cores only capture a very small area, and whether the derived melt records are representative should be corroborated with larger-scale stratigraphic analyses of the sub-surface, e.g. using multiple snow trenches or ground-penetrating radar.


Other general physical implications of melting for glaciers and glaciology include: Melt-freeze cycles change the thermal structure within the snowpack, i.e. (sub-)surface warming (Humphrey et al., 2012; Mattea et al., 2021), because freezing 1 g of water raises the temperature of 160 g adjacent snow by 1°C through the release of latent heat (Cuffey & Paterson, 2010). In extreme cases, this can lead to the development of highly permeable firn aquifers (Koenig et al., 2014; Kuipers-Munneke et

al., 2014; Miller et al., 2020). In addition, meltwater percolation, especially through PFPs, advects heat to deeper, sub-zero snow sections. At sites with preferential percolation, borehole temperatures can therefore be ambiguous (Pfeffer & Humphrey, 1996).

Since melt complicates firn densification through metamorphism rates differing between affected and unaffected sections, linear densification curve fits solely dependent on the overlying snow burden (Herron & Langway, 1980) are not sufficient to

predict internal density variations caused by melt. Improved understanding of firn meltwater retention (Vandecrux et al., 2020) and densification models (Ligtenberg et al., 2011) are therefore crucial, and differences in temporal resolution between unaffected firn and melt layers need to be taken into account when interpreting melt-affected ice cores (Moser et al., 2021).

Refrozen melt layers further reduce the permeability of the snow column, i.e. acting as impermeable barriers to percolation. The development of a stratigraphic boundary in turn encourages more ice aggregation at the top and/or bottom of the ice layer

and reinforces this effect. Though pneumatic testing has potential to uncover horizontal and vertical firn permeability in situ (Sommers et al., 2017), the effective permeability of refrozen melt layers for percolation is under debate. On the one hand, they can be partly permeable when warmed to melting point and allow for meltwater penetrating beyond these features (Lliboutry, 1996; Samimi et al., 2020). On the other hand, thick ice slabs are expected to be impermeable, because heat transfer in temperate, saturated firn is low (Samimi et al., 2020). Though the development of impermeable ice slabs leads to temporary

retention of melt, it also may reduce the firn's capabilities for meltwater storage (Vandecrux et al., 2019) and enhance runoff (Machguth et al., 2016; Pfeffer et al., 1991).

## 2.3. Identification and quantification of melt

### 2.3.1. Melt feature records from ice cores

In ice cores, melt events are commonly identified as visually distinct features. Distinction is either made manually during ice

core processing in the cold laboratory, where some melt layers can be clearly seen by eye or using line scanners (Brown et al., 2023; Dey et al., 2022; Kinnard et al., 2008; Svensson et al., 2005). Thus, long-term melt history records from Canada (Koerner



& Fisher, 1990), Greenland (Westhoff et al., 2022), Alaska (Winski et al., 2018), Arctic Russia (Fritzsche et al., 2005), and the Antarctic Peninsula (Abram et al., 2013) have been acquired.

Traditionally, the description of firn sections with melt features tends to be binary as sections 'affected' or 'unaffected' by melt. The resolution ranges between micrometres to a few millimetres, depending on sight or camera capabilities. An issue with this and other structural identification approaches is their inability to discriminate between very thin melt layers and other fine, high-density, bubble-free features like glaze (Albert et al., 2004; Fegyveresi et al., 2018; Frezzotti et al., 2002) and wind crusts (Sommer et al., 2018; Weinhart et al., 2021). While their discrimination requires further investigation, Westhoff et al. (2022), for example, applied a thickness threshold of 1.5 mm as upper limit for wind crusts and considered features >1.5 mm
melt-induced.

Furthermore, two melt section of equal thickness imply equal intensity of the melt events, even though various factors shape melt layer appearance (Sect. 2.1. & 2.2.). Thickness-based melt estimates don't reflect this complexity and have been shown to significantly overestimate melt proportion in comparison to melt constrained by line-scan derived density (Dey et al., 2022). Both visual processing and line scanning are limited to air bubble containing ice in the topmost few hundred metres of ice
above the bubble-to-clathrate transition zone (Neff, 2014), within which the air bubbles start to disappear as the air starts to form clathrates. Within this transition zone the visual differences between melt and non-melt sections (bubble density) vanish. However, this is generally not an issue for shallower, alpine ice cores and only relevant to deep polar ice cores.

Limitations inherent to the features are that (1) both rain-on-snow events and a non-accumulative positive surface energy balance can cause identical features in ice cores; (2) it remains unclear whether a thick melt feature with internal variability
resulted from one larger melt phase or several shorter events (Figure 1); (3) it has been shown that melt percolation of a single event can lead to several melt horizons (Westhoff et al., 2022). Given that ice cores are single-point records of highly dynamic and spatially heterogeneous environments (Zuhr et al., 2021), their representativity must be validated locally and obtained melt history records must be interpreted with caution. Quantitative melt estimates are commonly derived from the frequency and thickness of melt layers. Here, thinning of melt features due to burial-induced densification and glacier flow need to be taken
into account. A first melt parameter that does so is 'melt frequency' (MF), defined as number of melt events per depth (metre) or time unit (Alley & Anandakrishnan, 1995; Kaczmarska et al., 2006; Kinnard et al., 2008). This melt frequency can be used for calculating a second parameter, 'melt features percent' (MFP, Equation 2), i.e. the percentage of melt-affected ice ($M_{ice}$) of the annual layer (A) in terms of mass (Koerner, 1977a; Winski et al., 2018). The thickness-based approaches will lead to different profiles of the derived melt quantity (Supplementary Table 1).


$$MFP = M_{ice} \div A \times 100 \qquad \qquad \text{(Eq. 2)}$$

Besides the structural identification, chemical characteristics can be used to identify melt layers in ice cores. Methods based on the magnesium-sodium ratio (Li et al., 2006; Watanabe et al., 2015), chloride-sodium ratio (Eichler et al., 2001), trace-element concentration ratios (Avak et al., 2018), dissolved ion fraction percentage (Wu et al., 2018), and radionuclide content



(Pinglot et al., 2003) help to identify melt and constrain its effects. Rod-like inclusions (>30 µm) in ice layers have recently been suggested as indicator of large amounts of refrozen meltwater (Kawakami et al., 2022). Total air content, and in particular noble gas measurements have even proven valuable to detect melt where it cannot be detected visually (NEEM community members, 2013; Orsi et al., 2015).

**2.3.2. Modelling and remote sensing: potential and limitations for melt identification**

Various modelling and remote sensing approaches for the identification and quantification of melt exist, which can support ice core studies in the field. Hock (2005) have previously reviewed different types of melt models using the SEB. Such SEB models have great potential but are sometimes hampered by a lack of meteorological data beyond temperature. Simpler PDD models based on the relationship between melt and temperature (Krenke & Khodakov, 1966) are more commonly applied. However, they struggle to implement the small-scale spatial complexity of melt in heterogeneous topography as well as the 305 continuously changing meteorological and snowpack conditions, which hinders the detailed depiction of melt events.

Remote sensing has become a very influential way to assess melt occurrence in Antarctica, because it allows to investigate for large spatial coverage in the most remote region on Earth. Optical imagery (e.g. Landsat 8, Sentinel-2) and passive microwave sensors, which rely on the change of dielectric properties at the glacier surface (e.g. ICESat-2), are widely used today (Dell et al., 2021; Fricker et al., 2021). The short time span of such satellite observations (max. since 1980s) is a significant limitation 310 for studying changes in climate that take place on decadal to centennial time scales. Furthermore, the detection of surface melt events is directly dependent on the method sensitivity and chosen threshold, which can lead to the confusion of fresh melt events with retained subsurface liquid (Dell et al., 2021).

**3 Manifestations of melt in ice-core records**

**3.1. Structural characteristics of melt features in ice cores**

Melting leaves a physical imprint on the stratigraphy of the upper snowpack, including firn density, air content, and temperature. Here, we discuss the appearance of melt features in the firn stage, where structural differences are still visible, contrary to deep ice. Similar to glacier hydrology studies (Bell et al., 2018), which describe a variety of surface melt features on a macroscopic scale, it must be noted that traces of melt in ice cores can differ greatly, too.

A first common characteristic of melt-affected ice-core sections is their distinctly higher density caused by wet metamorphism 320 speeding up the compaction of the snowpack. This corresponds to lower porosity and permeability than the adjacent firn. Secondly, melt features generally contain few air bubbles (Das & Alley, 2005; Langway & Shoji, 1990; Orsi et al., 2015). While some studies state that melt layers are entirely bubble-free, some include bubble-sparse (Ahn et al., 2008; Koerner, 1989) or very coarse-grained yet bubble-containing sections in their definition of melt-affected ice (Macdonell et al., 2021). Depending on the speed of melt infiltration and freeze-off, the degree of degassing in the affected firn has been suggested to





vary, i.e. slower freeze off allows for more degassing and consequently fewer, smaller air bubbles in the melt-affected section (Ahn et al., 2008). Therefore, under rapid percolation, air from the open-porous firn can get trapped and merge to enlarged, irregular-shaped bubbles during the freezing process. These differ from regular air bubbles, unaffected by melt, which stem from long-term metamorphism and tend to have a more circular shape resulting from specific surface area reduction.

The impact of recrystallization on granulometry during melt phases has been used by Macdonell et al. (2021) to categorize

melt layers as fine (<1 mm), medium (1−2 mm), or large-grained (>3 mm). Another parameter to describe melt features in greater detail are their boundaries to the neighbouring firn. While many polar studies describe sharp contacts between non-melt and melt-affected sections (Das & Alley, 2005), irregular boundaries have also been observed (Fritzsche et al., 2005). Macdonell et al. (2021) observed 'diffuse, sinuous, and planar' contacts. The factors determining between homogeneous wetting and preferential flow, e.g. cold content and stratigraphy of the snowpack (Sect. 2.2), also drive melt layer prominence

(Culberg et al., 2021).

While melt layers are frequently reported as 'sub-horizontal' in ice cores with a limited sampling area, the different meltwater flow regimes discussed in Sect. 2.2 can lead to both horizontal and vertical features. Lateral flow is responsible for both discontinuous melt lenses and extensive melt layers (Das & Alley, 2005). Lateral connectivity of melt features is a useful parameter to describe the different degrees of melt impact on the snow stratigraphy, even at a glacier scale (Culberg et al.,

2021). At the same time, even when ice layers are continuous across a core, they cannot be assumed to be horizontally uniform (Kameda et al., 1995), and observations from a single ice core are not necessarily representative of the regional melting conditions.

Typical estimates for melt layer thickness are in the range of mm to dm (Das & Alley, 2005; Langway & Shoji, 1990; Orsi et al., 2015), but their extent varies according to the site-specific, short-term meteorological and snowpack conditions. At Site J,

southern Greenland, ~38% of melt layers were ≤2 mm thick (Kameda et al., 1995), with features up to 210 mm raising the mean melt thickness to ~10.8 mm. Thereby, 20–100 mm clusters of melt features are common within the ice cores from Site J (Kameda et al., 1995). In a Belukha ice core from the Siberian Altai, melt layers are mostly <10 mm thick with a few outliers >40 mm (Okamoto et al., 2011). Up to 85-mm thick ice layers have been recorded in an ice core from Mt. Yulong on the south-eastern Tibetan Plateau (Yuanqing et al., 2001). Melt features in the first sub-Antarctic ice core from Young Island are on

average 64 mm in thickness with a maximum of 580 mm (Moser et al., 2021; Thomas et al., 2021). At James Ross Island, Antarctic Peninsula, the mean vertical extent of melt layers is 32 mm (Abram et al., 2013).

These examples highlight two issues of melt layer records from ice cores: (1) whether close clusters of melt layers and interlayering of ice and firn are counted individually or collectively as one thicker melt-affected section is subjective to the processing team and can vary based on the eyesight or sample cut; (2) whether thick melt features are the result of a single

heavy melt event or a sequence of smaller melt events is indistinguishable during ice-core processing and requires further investigation of their internal stratigraphy.

Vertical melt features, so-called melt pipes or flow fingers, are direct evidence of percolation and distinct pathways for liquid beyond the 0°C wetting front (Pfeffer & Humphrey, 1998; Williams et al., 2010). Previous assessments of their diameter



summarized by Williams et al. (2010) range from ~36 mm (Marsh & Woo, 1984), generally 50−150 mm and ≤10 mm for low-
intensity events early in the season (McGurk & Kattelmann, 1988) to 10−400 mm (Campbell et al., 2006). Their spacing is
irregular with estimates ranging from every 30 to 60 mm to hundreds of millimetres in between. Due to their limited horizontal
extent, these features are hard to depict in ice cores with a diameter ≤10 cm. Furthermore, stratigraphic boundaries can cause
discontinuities of vertical flow features (Williams et al., 2010).

After all, the diverse appearance and structural characteristics of melt features in ice cores are not only influenced by the
meteorological and energy balance conditions at the glacier surface but strongly depend on the subsurface characteristics,
which determine meltwater flow (Fig. 3). Further investigation of commonalities and differences among melt features will
facilitate the development of a standardised description and identification scheme for melt-affected ice cores. This includes a
more detailed structural description of melt layers in ice cores from various climates including their natural variability, how to
handle the interplay of melt and firn, or how to interpret thicker melt layers seemingly resulting from several events.


**Figure 3: Examples of structural imprints of melt for the extreme scenarios of (a) cold firn and (b) temperate subsurface. Subsurface
temperatures influence the meltwater flow through the differing refreezing capacity of the snowpack and therefore can lead to
different melt features in firn. The profiles above show extreme examples of (a) cold firn with distinct melt layers, lenses and vertical
flow fingers, and (b) temperate snow characterized by more homogeneous progression of the melting front and wide-spread grain
coarsening. A continuum between these extremes allows for unique site- and event-specific structural manifestations of melt events
under natural conditions.**

### 3.2. Melt-affected chemical proxy records

Melting has diverse effects on ice-core records of, e.g., major ions, trace elements, organic compounds, or BC. Depending on
the chemical species, study site, and amount of melting, anything between a drastic disturbance and fully preserved records





has been reported in literature. For example, $NH_4^+$ (Avak et al., 2019; Trachsel et al., 2019), BC (Festi et al., 2021; Osmont et al., 2018; Pavlova et al., 2015), $^3H$ (Pinglot et al., 2003), low-abundant water-soluble and insoluble trace elements (Wong et al., 2013) appear to be rather preserved. On the other hand, melt-induced alteration is a substantial issue for $H_2O_2$ (Moser et al., 2021), major ions like $SO_4^{2-}$ and $Ca^{2+}$ (Eichler et al., 2001; Li et al., 2006; Virkkunen et al., 2007) and high-abundant water-

soluble trace elements (Avak et al., 2018, 2019).

Chemical impurities in the snow/firn part of a glacier can be, to varying degree, eluted from their original position of deposition and enriched either in a deeper layer during refreezing or in the meltwater run-off. Whether impurity species are displaced within or lost from a profile depends on the refreezing capacity of the subsurface, which can be anything between

predominantly cold firn to generally temperate snow and firn. Elution has been a research topic since the 1970s, because it leads to an environmentally relevant 'ion flush' from snowpacks at the start of the melt season (Costa & Pomeroy, 2019; Johannessen et al., 1977; Johannessen & Henriksen, 1978; Tsiouris et al., 1985). In this context, 'preferential elution' of certain ions compared to others has been observed, and a multitude of elution sequences for major ions has been published (Table 1 and references therein).


**Table 1: Chronological overview of studies on elution behaviour and sequences of major ions conducted in the field and laboratory using firn and ice-core sections, snow and meltwater composition measurements, and elution experiments. Elution sequences show strongly affected species left and more stable impurities to the right. The position of sulfate (bright pink), calcium (purple), chloride (dark blue) and ammonium (light blue) are highlighted in the elution sequences. Note that different study conditions and**

**presentation of elution in the referenced literature complicate a direct comparison.**

| Study Type | Location | Study | Publication Year | Elution Sequence |
|---|---|---|---|---|
| including firn and ice cores | Folgefonna, Norway | Davies et al. | 1982 | $SO_4^{2-}$, $Mg^{2+} > NO_3^- > Na^+$, $Cl^- > NH_4^+$, $K^+$ |
| | 18 C, Greenland | Goto-Azuma et al. | 1994 | $NO_3^- > SO_4^{2-} > Cl^-$ |
| | Ürümqi Glacier No.1, Tian Shan | Goto-Azuma et al. | 1994 | $NO_3^- > Mg^{2+} > SO_4^{2-}$, $Na^+ > Ca^{2+}$, $Cl^- > K^+$ |
| | Austre Brøggerbreen, Svalbard | Goto-Azuma et al. | 1994 | $NO_3^-$, $SO_4^{2-} > Na^+$, $Cl^-$ |
| | Upper Grenzgletscher, Switzerland | Eichler et al. | 2001 | $SO_4^{2-} > Ca^{2+}$, $Mg^{2+} > K^+$, $Na^+ >> NO_3^- > NH_4^+$, $F^- > Cl^-$ |
| | Lomonosovfonna, Svalbard | Moore et al. | 2005 | $NO_3^- > SO_4^{2-} > Ca^{2+}$, $MSA^-$, $K^+ > Cl^- > Na^+$, $NH_4^+ > Mg^{2+}$ |
| | Lomonosovfonna, Svalbard | Virkkunen et al. | 2007 | $SO_4^{2-}$, $Ca^{2+}$, $Mg^{2+}$, $NO_3^- > K^+ > Cl^- > Na^+ > NH_4^+$ |
| | Nevado Coropuna Saddle, Peru | Herreros et al. | 2009 | $SO_4^{2-} > NO_3^- > Mg^{2+} > NH_4^+ > Ca^{2+} > Na^+ > K^+ > Cl^- > Br-$ |
| | Lomonosovfonna snow pit, Svalbard | Moore and Grinsted | 2009 | $Ca^{2+}$, $SO_4^{2-}$, $Mg^{2+}$, $NO_3^- > K^+ > Cl^- > Na^+ > NH_4^+$ |
| | Lomonosovfonna ice core, Svalbard | Moore and Grinsted | 2009 | $SO_4^{2-}$, $NO_3^- > Mg^{2+} > Ca^{2+} > Cl^- > Na^+ > K^+ > NH_4^+$ |
| | Austre Brøggerbreen, Svalbard | Moore and Grinsted | 2009 | $Mg^{2+} > SO_4^{2-} > NO_3^- > Ca^{2+} > Cl^- > Na^+ > K^+ > NH_4^+$ |
| | Chimborazo, Ecuador | Ginot et al. | 2010 | $SO_4^{2-} > Ca^{2+} > Mg^{2+} > F^- > Na^+$, $K^+ > Cl^- > NO_3^- > NH_4^+$, $HCOO^-$ |
| | Lomonosovfonna, Svalbard | Vega et al. | 2016 | $NO_3^- > SO_4^{2-} > Mg^{2+}$, $Cl^-$, $K^+$, $Na^+$ |
| | Grand Combin, Switzerland | Huber et al. | 2020 | $MSA^- > SO_4^{2-}$, $C_2O_4^-$, $Ca^{2+} > NO_3^-$, $Mg^{2+} > F^-$, $NH_4^+$ |
| Measurements of snow and/or | Cairn Gorm Mountain, Scotland | Brimblecombe et al. | 1985 | $SO_4^{2-} > NO_3^- > NH_4^+ > K^+ > Ca^{2+} > Mg^{2+} > H^+ > Na^+ > Cl^-$ |
| | Ürümqi Glacier No.1, Tian Shan | Li et al. | 2006 | $SO_4^{2-} > Ca^{2+} > Na^+ > NO_3^- > Cl^- > K^+ > Mg^{2+} > NH_4^+$ |



| | Location | Reference | Year | Order |
|---|---|---|---|---|
| | Holtedahlfonna, Spitsbergen | Virkkunen et al. | 2007 | $SO_4^{2-}$, $Ca^{2+}$, $Mg^{2+}$, $NO_3^- > K^+ > Cl^- > Na^+ > NH_4^+$ |
| | Baishui Glacier No.1, Tibetian Plateau | Pang et al. | 2012 | $NO_3^- > SO_4^{2-} > K^+ > Cl^- > Mg^{2+} > Ca^{2+} > Na^+$ (profile 1) |
| | Baishui Glacier No.1, Tibetian Plateau | Pang et al. | 2012 | $K^+ > SO_4^{2-} > NO_3^- > Cl^- > Ca^{2+} > Mg^{2+} > Na^+$ (profile 2) |
| | Midtre Lovénbreen, Svalbard | Björkman et al. | 2014 | $SO_4^{2-}$, $Cl^- > NO_3^-$, $Na^+$, $K^+ > Mg^{2+} > Ca^{2+} > F^-$, $NH_4^+ >$ microbes |
| | Ürümqi Glacier No.1, Tian Shan | You et al. | 2015 | $SO_4^{2-} \gg Mg^{2+}$ |
| | Shiyi Glacier, Qilian Mountains | Zong-Xing et al. | 2015 | $SO_4^{2-} > K^+ > Ca^{2+} > Mg^{2+} > Na^+ > Cl^- > NO_3^-$ |
| | Lomonosovfonna, Svalbard | Vega et al. | 2016 | $NO_3^- > SO_4^{2-} > Mg^{2+}$, $Cl^-$, $K^+$, $Na^+$ |
| | Tapado Glacier, Chile | Sinclair and MacDonell | 2016 | $SO_4^{2-}$, $Na^+$, $Mg^{2+}$, $Ca^{2+} \gg Cl^-$ |
| | Ürümqi Glacier No.1, Tian Shan | Wu et al. | 2018 | $Ca^{2+} > Na^+ > Mg^{2+} > K^+$ |
| | Baishui Glacier No.1, Tibetian Plateau | Wang et al. | 2018 | $Ca^{2+} > SO_4^{2-} > NH_4^+ > K^+ > NO_3^- > Na^+ > Cl^- > Mg^{2+}$ |
| | Weißfluhjoch, Switzerland | Avak et al. | 2019 | $SO_4^{2-}$, $Ca^{2+}$, $NO_3^- > Na^+$, $Cl^- > NH_4^+$ |
| | Weißfluhjoch, Switzerland | Trachsel et al. | 2019 | $Ca^{2+}$, $SO_4^{2-}$, $Na^+ > NH_4^+$, $F^-$, $Cl^-$ |
| | Austre Brøggerbreen, Svalbard | Spolaor et al. | 2021 | $Na^+ > I^- > Br^- > K^+ > Mg^{2+} > Ca^{2+} > Cl^-$ (melt events) |
| | Austre Brøggerbreen, Svalbard | Spolaor et al. | 2021 | $Mg^{2+} > Na^+ > K^+ > SO_4^{2-} > Ca^{2+} > I^- > Cl^- > MSA^-$ (rain on snow) |
| Elution experiments | Southeastern Norway | Johannessen et al. | 1977 | $SO_4^{2-} > Ca^{2+} > Mg^{2+} > H^+ > NH_4^+ > Cl^- > NO_3^- > Na^+ > K^+$ |
| | Folgefonna, Norway | Tsiouris et al. | 1985 | $Ca^{2+} > NO_3^- > K^+ > Mg^{2+} > SO_4^{2-} > Na^+ > Cl^-$ |
| | — | Hewitt et al. | 1989 | $SO_4^{2-} > Ca^{2+}$, $Mg^{2+} > K^+ > Na^+ > NO_3^- > Cl^-$ |
| | — | Tranter et al. | 1992 | $SO_4^{2-} > Mg^{2+} > Na^+$, $NO_3^- > Cl^-$ |
| | Sasagamine, Niigata, Japan | Goto-Azuma et al. | 1993 | $NO_3^- > SO_4^{2-} > Cl^-$ |
| | — | Herrmann and Kranz | 1995 | Alkali metals, alkaline earth metals, cations (other than $NH_4^+$) > $SO_4^{2-} > NO_3^- > Cl^- > NH_4^+ > H_2O_2$ |
| | — | Cragin et al. | 1996 | $SO_4^{2-}$, $NO_3^- > \ldots > Cl^-$ |
| | — | Costa and Pomeroy | 2019 | $PO_4^{3-} > NO_3^- > Cl^- > SO_4^{2-} > F^-$ |
| Review | — | Kuhn | 2001 | $SO_4^{2-} > NO_3^- > Cl^-$ |





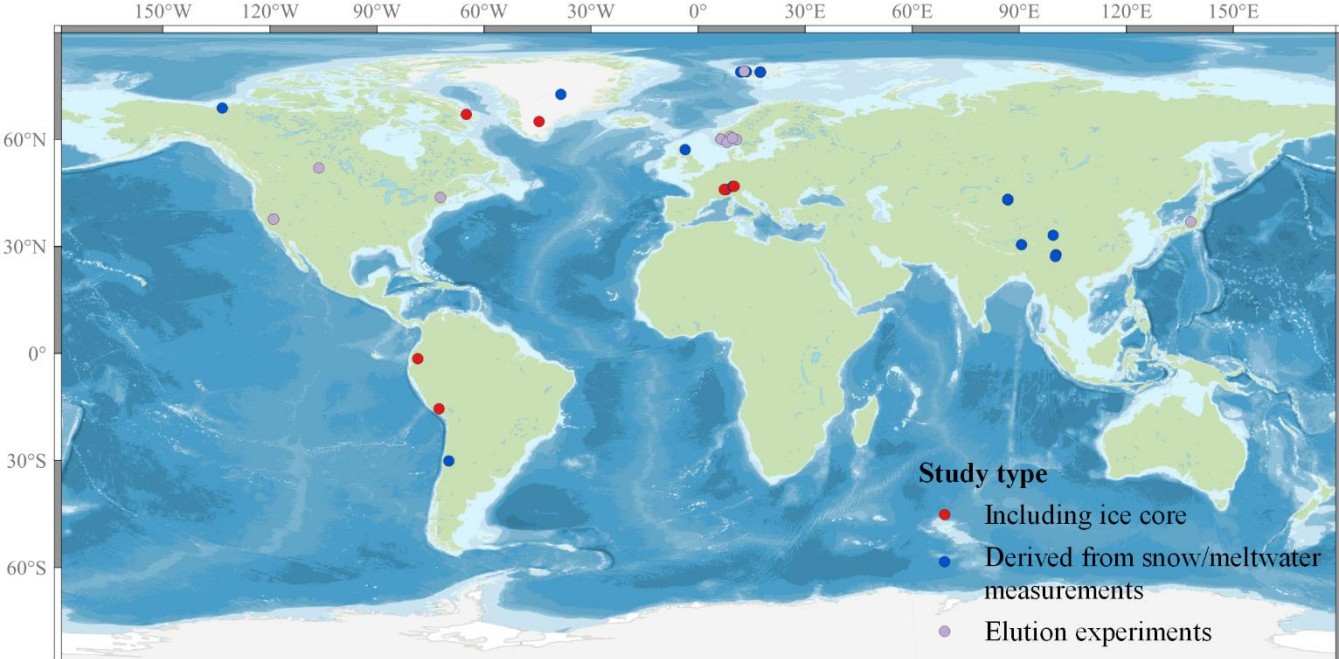

**Figure 4: Overview of study sites, where the effects of melting on major ion chemistry has been conducted using ice cores (red), snow and meltwater measurements in the field (blue) and elution experiments in the laboratory and field (light purple).**


The elution sequences presented in Table 1 have been obtained from natural and synthetic snow and ice samples, in field and laboratory experiments, e.g. lysimeter measurements, at various sites as well as from modelling approaches (Figure 4). Related to the different sites and study conditions, and to the sample material of varying age and composition, there is partial disagreement in the exact position of ions in the sequence. Nevertheless, sulfate and calcium tend to be among the ions eluted

more quickly, while fluoride, chloride and ammonium appear more stable (Table 1, Fig. 5). Since snow has been shown to reveal no chromatographic properties, this is not due to a potential retention of latter species in the snow/firn with respect to former ones during elution (Cragin et al., 1996; Hewitt et al., 1989; Tranter et al., 1986). Instead, several other factors shaping the elution sequences of different major ions and also other chemical species have been identified.



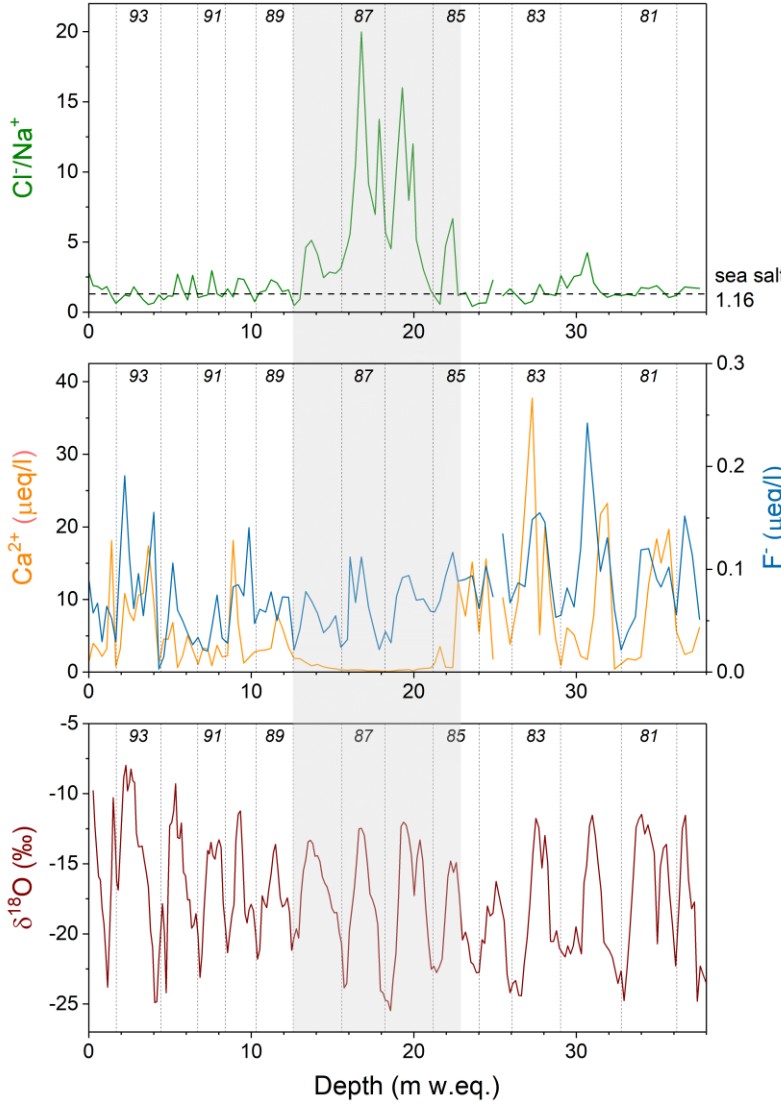


**Figure 5: Records of Cl⁻/Na⁺ ratio (green), Ca²⁺ (orange) and F⁻ (blue) concentrations, and δ¹⁸O (brown) within the upper 38 m w.eq. of an ice core from Grenzgletscher (Swiss Alps, Eichler et al., 2001), covering the time period 1980–1994 (individual years are indicated, dotted lines): Meltwater inflow through a crevasse system led to (a) high Cl⁻/Na⁺ ratios, exceeding the sea-salt ratio (1.16, dashed line) in the melt-disturbed range (grey background), (b) preferential**

**elution of Ca²⁺ compared to F⁻, with the latter still showing well-preserved concentration seasonality in the depth range ~12–23 m w.eq., (c) whereas δ¹⁸O is mainly undisturbed; observations suggest lack of refreezing of the percolating meltwater but drainage along the firn-ice transition.**

The primary determining factor for the behaviour of impurities during elution is their solubility in water (Meyer and Wania,

2011; Fig. 6) and ice (Avak et al., 2019). A high solubility in ice, e.g. observed for F⁻, Cl⁻, $NH_4^+$, allows for chemical species

to be incorporated into the ice lattice during snow metamorphism and thereby reduces their availability to meltwater elution



(Cragin et al., 1996; Trachsel et al., 2019; Eichler et al., 2001), while high solubility in water enhances their mobility during melt events. Insoluble microparticles are less affected by displacement to deeper layers of the snowpack and can show melt-induced peaks at former surfaces (Koerner, 1977b, 1997). Meyer and Wania (2011) have modelled the elution behaviour of
organic compounds from a homogeneous snowpack and distinguish four elution types (Figure 6): (a) impurities strongly soluble in water are released quickly after melt onset; (b) particulate matter and surface-adsorbed impurities elute last; (c) meltwater concentrations of species that are partly water-soluble and partly sorbed to ice grain surfaces increase gradually with melt duration; (d) bimodal impurities that can behave both like dissolved type (a) and particulate type (b) show peak concentrations at the beginning and end of the snowmelt season.

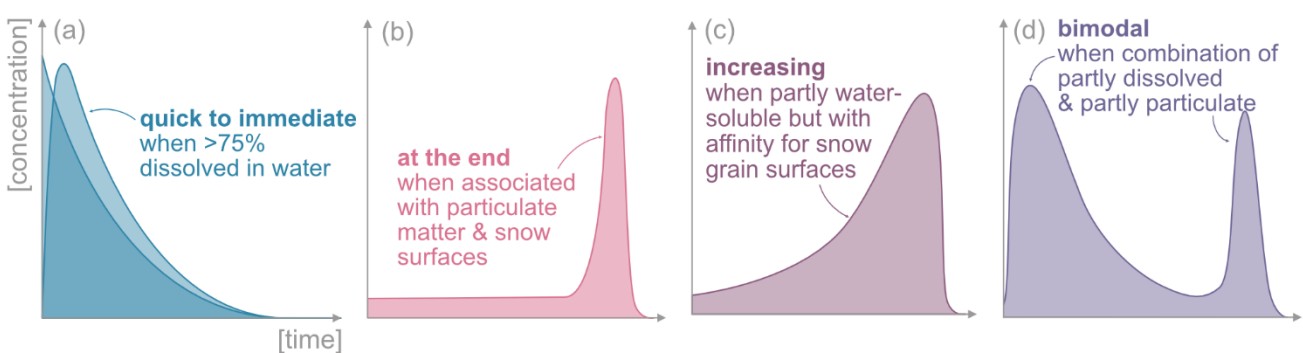


**Figure 6: Differences in behaviour during melt elution of (a) water-soluble impurities, (b) particle-bound matter, (c) species that are partly water-soluble and partly sorbed to ice grain surfaces, and (d) impurities combining the dissolved type (a) and particulate type (b); this is a schematic summary based on modelling of organic compounds during melt by** Meyer and Wania (2011)**.**

Besides their solubility in water and ice, (1) pre-melting snow characteristics, (2) snow metamorphism, and (3) the character of the melt process itself can subordinately influence the elution behaviour of impurities (Fig. 7).

 (1) Initially, precipitation formation or atmospheric scavenging processes influence the chemical composition of the snow and its absolute ion concentrations and thus, the impurity behaviour during elution. The initial position of impurities within the ice-air matrix depends on whether they act as condensation nuclei during snow crystal formation (placed within
crystal) or are scavenged and found in the exterior of snow grains (Trachsel et al., 2019; Tsiouris et al., 1985). However, even ions that have been deposited together like $Na^+$ and $Cl^-$ (sea-salt aerosol) showed partially different behaviour during leaching (Eichler et al., 2001), and metamorphism subdues this factor post-depositionally. Since only a certain amount of impurities can be incorporated into the ice lattice, absolute concentrations affect the number of ions placed at grain boundaries, where they are more available to interaction with meltwater. Less abundant (partially) water-soluble trace
elements appear to be more stable than the ones with higher absolute concentrations (Avak et al., 2018, 2019; Li et al., 2006).



(2) The physical location of impurities within the ice matrix is relevant beyond the initial position and undergoes changes with ageing during snow metamorphism (Kuhn, 2001; Moore et al., 2005; Pohjola et al., 2002; Trachsel et al., 2019; Tsiouris et al., 1985). Earlier studies suggest that diurnal melt-freeze cycles generally enhance the availability of ions to leaching during percolation (Johannessen and Henriksen, 1978; Goto-Azuma et al., 1994). However, it has been shown that rearrangement behaviour during metamorphism differs among impurities (Cragin et al., 1996; Hewitt et al., 1989; Trachsel et al., 2019) mainly depending on their solubility in ice (Avak et al., 2019; Eichler et al., 2001). For ions, this solubility in ice is influenced by their size and charge (Moore & Grinsted, 2009) but also by the capability to form volatile compounds, which is e.g. significant for $Cl^-$, $F^-$, $NO_3^-$, and $NH_4^+$ (Eichler et al., 2001; Li et al., 2006). On this basis, $Cl^-$, $NH_4^+$ and $F^-$ tend to be incorporated in the ice matrix during advancing metamorphism (Eichler et al., 2001; Ginot et al., 2010; Moore et al., 2005; Trachsel et al., 2019), whereas $SO_4^{2-}$ and $Ca^{2+}$ tend to be excluded and are more readily available to meltwater relocation. The effect of metamorphism on the preservation or loss of ions in the snowpack becomes increasingly apparent with longer duration of metamorphism (Brimblecombe et al., 1987; Cragin et al., 1996; Trachsel et al., 2019).

Furthermore, the temperature-gradient metamorphism conditions, especially absolute (diurnal mean) air temperatures and their diurnal variability (Li et al., 2006; Virkkunen et al., 2007), determine between constructive and deconstructive metamorphism, influence the near-surface snow grain shape, and indirectly constrain the availability of impurities at grain boundaries through the snow surface area. Constructive metamorphism leads to hoar and faceted crystals with a higher specific surface area and the solubility-driven separation between interior and exterior of a snow grain is less effective than during deconstructive metamorphism in rounded grains (Cragin et al., 1996). A temperature gradient is generally necessary to achieve recrystallisation rates that lead to significant chemical re-arrangement (Trachsel et al., 2019).

(3) Differences in meltwater flow type, melt rate, addition of rain on snow, and snow structure make the character of the melt event itself a shaping factor for elution behaviour of ions. Costa and Pomeroy (2019) documented that preferential flow paths speed up the release of $SO_4^{2-}$ and $NO_3^-$ from the snowpack. A slower melt rate leads to higher ionic pulses (Davis et al., 1995), because it allows for longer interaction with and leaching of the snow. Similarly, melt events without rain on snow appear to be more efficient in eluting $SO_4^{2-}$ and $NO_3^-$, because PFPs developing under these circumstances result from microscale flow structures (Costa & Pomeroy, 2019; Tranter et al., 1992) that differ from hydrologically constrained PFPs during rain-on-snow events. Furthermore, the snow texture (e.g. specific surface area, SSA) strongly affects hydrological permeability, and less or im-/permeable layers can lead to the retaining and refreezing of ion-enriched meltwater within the profile (Pfeffer & Humphrey, 1996).



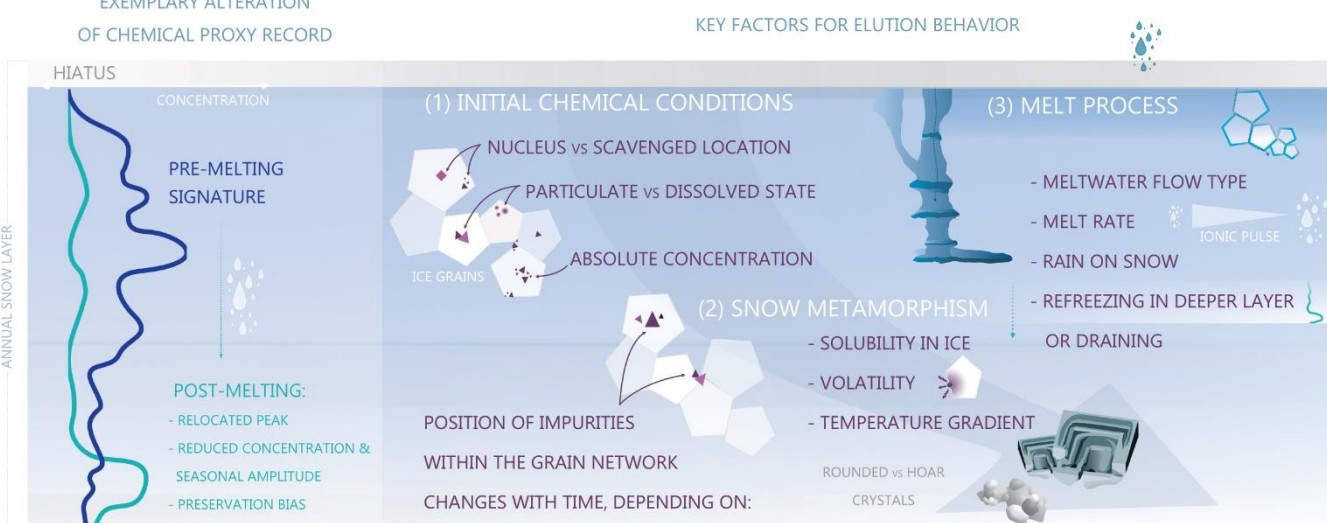

**Figure 7: Schematic display of issues associated with melt-affected ice-core chemistry using a qualitative drawing (left), and key factors for elution behaviour of impurities (right) include (1) initial chemical conditions of the deposited snow, (2) segregation of impurities during snow metamorphism, and (3) the melt process itself.**

Melt-induced alteration of chemical ice-core proxy records have further been assessed using ion elution factors (Moore & Grinsted, 2009), ion ratios like $Mg^{2+}/Na^+$ (Iizuka et al., 2002; Watanabe et al., 2015) and $Cl^-/Na^+$ (e.g. Fig. 5, Eichler et al., 2001), and based on the dissolved fraction percentage (DFP, Wu et al., 2018; Zhang et al., 2013). Comparing concentrations of a chemical species in a snow layer before and after a melt event is the basis of elution factor quantification (Moore & Grinsted, 2009). The ion ratio approaches indicate melt-affected profile sections based on the differing elution factor of two ions like $Mg^{2+}$ and $Na^+$, which have a common source at the investigated site. The DFP is defined as:

$$DFP = X_{dis} / (X_{dis} + X_{insol}) \times 100\% \qquad\qquad (Eq.\ 3)$$

Changes in the preserved ions-to-microparticles ratio of soluble ($X_{dis}$) and insoluble fraction ($X_{insol}$) as formulated in Equation 3 reflect the intensity of post-depositional alteration through melting when source and composition of the impurities are homogeneous in a layer pre-melting (Zhang et al., 2013). When compared against DFP values of non-melt affected parts from the same site, lowered DFP signatures reflect the mobility of dissolved and immobility of insoluble impurities during elution (Fig. 7). Therefore, site-specific influences on the behaviour of chemical species used for $Mg^{2+}/Na^+$, $Cl^-/Na^+$ ion ratios and DFP must be considered. While elution factor and $Mg^{2+}/Na^+$ or $Cl^-/Na^+$ ratio calculations are easier in their application, the DFP has the advantage that it can identify gradual stages of elution.

Finally, the questions remain, to which extent and in which way climate and environmental information are still preserved in melt-affected chemical records. Firstly, dating melt-affected ice cores has proven challenging in locations ranging from alpine





(Festi et al., 2021; Pavlova et al., 2015) through to sub-Antarctic sites like Young Island (Moser et al., 2021). In this context, the melt-induced hiatus in the ice-core layering during melt events (Koerner, 1997) is a major issue. If sufficient summer precipitation is altered or removed (Yuanqing et al., 2001), it can lead to a seasonal bias of the net annual layer towards winter

accumulation. In addition, the resolution in melt-affected chemical records can be lowered from seasonal to (multi-)annual (Moore & Grinsted, 2009; Vega et al., 2016) due to a reduction of soluble ion concentrations and of the amplitudes of seasonal proxies like hydrogen peroxide (Davies et al., 1982; Huber et al., 2020; Moser et al., 2021). Consequently, annual layer counting as a dating approach can be severely hampered in melt-affected soluble impurity records. However, using chemical species less affected by meltwater percolation such as $NH_4^+$, insoluble trace elements and particulate records like BC has

already proven valuable for dating melt-affected ice (Avak et al., 2019; Neff et al., 2012; Pavlova et al., 2015). With regard to more confined dating horizons, $^{137}Cs$ cannot be used due to its melt-induced relocation, but $^3H$ (Pinglot et al., 2003; Van Der Wel et al., 2011) and $^{210}Pb$ (Gäggeler et al., 2020; Neff et al., 2012) remain viable as dating tools.

Both (1) percolation extent and (2) species-dependent elution behaviour must be considered in order to decide which melt-
affected chemical impurity records can still be used for environmental reconstructions, and how they can be used.

(1) Elution length determines the preserved resolution of melt-affected chemical impurity records. Annual means of environmental proxies remain suitable where melt is seasonally confined. For example, this is the case at Lomonosovfonna in Svalbard (Moore et al., 2005; Pohjola et al., 2002; Vega et al., 2016), Penny Ice Cap (Grumet et al., 1998) and Combatant Col at Mount Waddington in Canada (Neff et al., 2012). If percolation affects more than

one annual layer, multi-year averages are more reliable, e.g. at Severnaya Zemlya in the Eurasian Arctic (Weiler et al., 2005), Holtedahlfonna (Beaudon et al., 2013), or Austfonna in Svalbard (Tarussov, 1992).

(2) Among major ions, $NH_4^+$, $F^-$ and $Cl^-$ are generally better preserved than $Ca^{2+}$, $SO_4^{2-}$ (Table 1). Thus, $NH_4^+$ is preferred to $SO_4^{2-}$ for reconstructing anthropogenic changes, and $Cl^-$ preferred to $Na^+$ for investigating sea-salt changes. For trace elements (TEs), it has been shown that insoluble, particle-bound TEs are rather immobile (Avak et al., 2018,

2019; Gabrieli et al., 2011; Wong et al., 2013), whereas relocation of water-soluble TEs depends on their absolute concentration levels, with ultra-TEs better preserved. Thus, past changes in dust input can be established more reliably using rare earth elements (REEs) instead of Ba, Mg or Ca records. Pb, Cu, Sb, and Ag are still applicable for anthropogenic reconstructions, whereas Cd and Zn are less reliable proxies. Looking at organic species, the few available studies show widely differing responses to meltwater input depending on whether the organic material

consists of dissolved, hydrophobic or adsorbed species (Meyer & Wania, 2011; Schöndorf & Herrmann, 1987; Simmleit et al., 1986). For example, polychlorinated biphenyls lean towards relocation (Pavlova et al., 2015), and out of selected mono- and dicarboxylic acids and α-carbonyls, only methylglyoxal showed potential as a better-preserved organic aerosol proxy in melt-affected records (Müller-Tautges et al., 2016). Regarding particulate impurities like BC (and pollen), preserved annual cycles in temperate glaciers suggest that BC and pollen can still provide an important

environmental proxy even under strong melting conditions (Festi et al., 2021; Pavlova et al., 2015; Takeuchi et al.,





2019). Consequently, BC appears to be more suitable, e.g., for reconstructing the forest-fire history compared to organic fire proxies prone to leaching such as *P*-hydroxybenzoic acid (Müller-Tautges et al., 2016).

It must be noted that the observed meltwater persistence of certain species at one location is not directly transferrable to ice-core archives from other sites (e.g. Table 1). Since the impurity response to post-depositional melting processes is shaped by numerous factors like impurity concentrations, chemical composition, mineralogy, or amount of meltwater, it may vary between remote and polluted regions, different climate conditions, or with the distance to deserts and oceans. Thus, the preservation of chemical species, in each case, requires site-specific assessment.

### 3.3. Stable water isotopic (SWI) signatures affected by melt

Isotopic fractionation is fundamental to the interpretation of stable water isotopes (SWI, Gat, 1996) as climate proxies in ice cores. Since the fractionation of heavier and lighter isotopes takes place during phase changes, melting and refreezing in the snowpack are a significant factor (Koerner et al., 1973). Due to isotopic exchange between liquid water and surrounding ice matrix, the linear slope between the abundance of $\delta^{18}$O and $\delta^2$H is considered altered (Ham et al., 2019; Lee et al., 2020). While normally displaying a slope of 8, it is often lowered in melt-affected ice (Souchez & Jouzel, 1984). For this reason, 555    melt-affected SWI signatures have been considered biased (Koerner et al., 1973; Moran & Marshall, 2009), i.e. enriched in heavy isotopes $^{18}$O and $^2$H. The presence of liquid water further leads to smoothing of the stable water isotopic records in snow pits and ice cores (Kang et al., 2008; Matsuoka & Naruse, 1999; Nye, 1998), "with the degree of smoothing dependent on the amount of melting and percolation" (Arnason et al., 1973; Pohjola et al., 2002). Another effect of percolation frequently discussed is the redistribution of isotopic signatures into deeper layers, which causes the mixing of signatures from different 560    seasons (Koerner, 1997). Both altered fractionation and signal percolation can lead to an amplitude reduction or entire loss of the seasonal cycle preserved in stable water isotope ratios like $\delta^{18}$O (Moran & Marshall, 2009; Rempel et al., 2001; Shiraiwa et al., 2002; Yuanqing et al., 2001).

The preservation of deposited isotopic signatures depends on the interplay of various factors and remains a subject of research. 565    If percolation is mainly pore flow and leaves the ice matrix intact, for example when the liquid-ice contact time is short, the SWI signal is less affected. Percolation depth is another important factor for $\delta^{18}$O preservation, as melting and refreezing limited to the surface leads to distinct ice layers (Sect. 2.2. and 3.1.), which can act as impermeable barriers for diffusion caused by movement of water vapour (Van Der Wel et al., 2011). The study argues that ice cores affected by short-term melt and refreezing events, which divide the snow column into separate pore spaces, could exhibit more reliable stable water isotope 570    records, because water vapour diffusion between the spaces is prohibited.

Several studies from SWI records at melt-affected sites proved to reliably record past temperature changes with seasonal resolution, e.g. in the Swiss Alps (Eichler et al., 2001; Fig. 6) and Patagonia (Schwikowski et al., 2013), or at annual resolution,




e.g. in Canada (Neff et al., 2012), Svalbard (Pohjola et al., 2002; Van Der Wel et al., 2011), and Siberian Altai (Okamoto et
al., 2011). Where percolation is not restricted to annual layers, multi-year averages could still be used. For example, long-term
(>5 years) climate variability is currently still preserved at Holtedahlfonna, Svalbard, despite seasonal variability being lost
(Spolaor et al., 2023). When considering millennial-scale features, the $\delta^{18}$O ratios in a melt-affected part of the North Greenland
Eemian Ice Drilling ('NEEM') ice core seem unaffected (NEEM community members, 2013). On the contrary, at Quelccaya
Ice Cap in Peru, strong meltwater-related smoothing of the annual signal and an increase of the SWI ratios between two cores
drilled in 1976 and 1991 has been observed, rendering the reconstruction of past temperature from SWI ratios impossible at
this site (Thompson et al., 1993). Similarly severe smoothing issues and loss of $\delta^{18}$O as dating tool and climate proxy have
been reported at Hielo Patagónico Norte, South America (Matsuoka & Naruse, 1999), and numerous other low-latitude high-
altitude glaciers (Thompson et al., 2021, and references therein). The recent study by Spolaor et al., 2023) highlights how
intensifying melt has gradually degraded the SWI records from Holtedahlfonna, Svalbard, over the course of less than a decade.
Thus, the reliability of SWI records from melt-affected ice cores as climate proxies is strongly dependent on the study site and
progressing melt conditions.

### 3.4. Gas records affected by melt

Atmospheric air entering the snow at the surface is diffusing through the snow and firn reaching the bottom of the firn column
much faster than snow of the same age. In the so-called lock-in-zone, the pore space within the firn closes off, thereby trapping
the air at 50–100 m below the snow surface depending on local conditions (Neff, 2014). This leads to an absolute age difference
between ice and gas phase commonly referred to as delta age (Δage; Schwander and Stauffer, 1984), which depends on the
vertical compaction of the firn column, mainly a function of snow accumulation at the surface, and can range from decades
for higher accumulation sites to many millennia for the East Antarctic plateau (Bazin et al., 2013; Buizert et al., 2015; Roberts
et al., 2017; Veres et al., 2013).

In contrast to the chemical parameters discussed in Sect. 3.3., samples of atmospheric air trapped in tiny bubbles in the ice
matrix are only indirectly affected by melt. The influence of melt layers on the air movement in the firn column and bubble
enclosure characteristics remains poorly understood (Keegan et al., 2014, 2019; Trudinger et al., 1997). Assuming that melt
layers behave similar to (very) dense layers of firn, published research on gas diffusion in non-melt affected firn suggests:

(1) Discontinuous melt layers may act as a vertical barrier and enhance horizontal air flow, thereby potentially reducing
the effects of gravitational settling (Birner et al., 2018; Craig et al., 1988). However, in non-extreme cases melt lenses
likely cannot completely prohibit air diffusion.

(2) Frequent melt layers may leave a lasting effect on firn metamorphism (Keegan et al., 2019) potentially changing the
pore volume and increasing the height of the lock-in-zone. The latter would cause younger air to be trapped earlier,
leading to a decrease in the mean age of the trapped air, i.e. it may widen the gas age distribution in closed bubbles
(Mitchell et al., 2015).



Focussing on the gases contained, melt layers differ from non-melt-affected ice found below the lock-in zone in three important ways. Most importantly, air is dissolved in the meltwater following Henry's law. Depending on the speed of refreezing, not
all of these gases will degas, and some fraction will remain trapped in the refrozen melt feature. The relative amount fractions of key gas species used for palaeoclimatic reconstructions, including carbon dioxide ($CO_2$), methane ($CH_4$), and noble gases Argon (Ar), Krypton (Kr) and Xenon (Xe), increase by differing amounts relative to the atmospheric levels in liquid water (Ahn et al., 2008; Neftel et al., 1983; Orsi et al., 2015). In a melt layer, the $CO_2$ amount fraction may increase by up to two orders of magnitude relative to the atmospheric amount fraction with equilibrium levels at ≥16,000 ppm at 0°C (Ahn et al.,
2008; Weiss, 1974), so even trace amounts of melt have the potential to significantly alter atmospheric $CO_2$ reconstructions. Note that over centuries to many millennia, the gas trapped in the melt layer will slowly diffuse into the surrounding layers of firn/ice driven by a large concentration gradient and alter the composition of trapped air once the melt layer passed the lock-in zone, thereby also altering the gas composition in the surrounding ice on the cm-scale (Ahn et al., 2008). The same issue of gas dissolution is encountered during wet extraction commonly used for $CH_4$ amount fraction measurements and requires a
dissolution correction, while the dry-extraction or sublimation methods typically used for $CO_2$ measurements prevents this as no water is present in these cases. The absolute gas concentrations in melt-affected profiles depend on (1) the speed at which melt is refreezing, (2) gas equilibration while passing through the firn column, and (3) the amount of atmospheric air trapped in the melt-layers during bubble enclosure (Stauffer et al., 1985).

Secondly, the presence of liquid water may trigger the production of $CO_2$, $CH_4$ and nitrous oxide ($N_2O$) of non-atmospheric origin in the ice, either by acid-carbonate reaction (Ahn et al., 2008; Delmas, 1993; Orsi et al., 2015; Tschumi & Stauffer, 2000), oxidation or biodegradation of organic compounds (Lee et al., 2020; Tschumi & Stauffer, 2000), or microbial production (Lee et al., 2020; NEEM community members, 2013). Together with the solubility-related increase of greenhouse gases, amount fractions as high as 2000 ppm $CO_2$ (Ahn et al., 2008; Neftel et al., 1983), 1200 ppb $CH_4$, and 150 ppb $N_2O$
(NEEM community members, 2013) above background levels for Greenlandic ice cores have been reported. The production potential for Antarctic ice core is much lower due to overall lower dust deposition, but still sufficient to significantly alter the atmospheric amount fractions (Ahn et al., 2008).

Lastly, total air content (TAC) is normally interpreted as driven by the elevation of the glacier surface (Delmotte et al., 1999;
Martinerie et al., 1992). However, a reduced TAC can also stem from percolation and refreezing of meltwater as melt layers, where it may be reduced by up to about 40% (Stauffer et al., 1985). This reduction does not seem to be consistent throughout different melt features and ice cores. TAC within a melt layer likely depends on the speed at which the meltwater refreezes, with faster rates being associated with higher TAC due to less time available for degassing of the gas solved in the water phase (Ahn et al., 2008; NEEM community members, 2013; Orsi et al., 2015). Observations from the Greenland NEEM ice core




during the Eemian Interglacial show that TAC varies more strongly during warm phases with more frequent melt events
(NEEM community members, 2013).

The effects described above can be used to identify and quantify depth sections affected by melt. While the presence of a melt
layer may lead to a major increase in greenhouse gas amount fractions, excessive amounts of greenhouse gases alone are not
sufficient to identify melt, because of the potential for in-situ production in the absence of liquid water (Schilt, Baumgartner,
Blunier, et al., 2010; Tschumi & Stauffer, 2000). Noble gases, on the other hand, are not produced in-situ. The solubility of
the noble gases Kr and Xe are approximately two and four times greater than for Ar, which has proven to be a sensitive tool
to confirm suspected melt in deep sections of the Greenland NEEM ice core (Orsi et al., 2015). Orsi et al. (2015) showed that
$\delta(Kr/Ar)$ and $\delta(Xe/Ar)$ of melt-affected depth sections, 44±12‰ and 109±34‰ (n=6), were significantly higher than
background levels of unaffected ice, 13±2‰ and 24 ±4‰. Based on only two samples, Ahn et al. (2008) reported even greater
sensitives for $\delta(Kr/Ar)$ and $\delta(Xe/Ar)$ of melt-affected section in excess of 100‰ and 250‰, respectively. If the presence of
even small amounts of melt is suspected, a pre-emptive scan of the ice for hidden melt is a sensitive tool. Unfortunately, the
necessity for preparing discrete samples, large ice sample requirements (typically 2-3x 50-60 g), and a limited daily sample
throughput of the noble gas method (Orsi et al., 2015) do not allow for a systematic implementation. It should be noted that
the noble gas method cannot quantify the melt intensity.

Aside from noble gases, low TAC could be interpreted as an indicator for melt, as has been suggested based on TAC simulation
(Plach et al., 2021; Vudayagiri et al., 2018). However, this method requires evidence of a significant TAC decrease for visible
melt layers for the studied ice core, and the TAC for melt layers may be very similar to non-melt affected ice (Orsi et al.,
2015).


So far, only two lower-elevation Antarctic ice cores with small amounts of melt have been used for gas reconstructions. Firstly,
the Law Dome ice core, which was drilled on an isolated dome at 1370 m above sea level close to the continental edge of East
Antarctica and contains ≤5 melt layers (Etheridge et al., 1996; Trudinger et al., 1997). Secondly, the Siple Dome ice core,
which is located on the West Antarctic Ice Sheet at an altitude of 620 m and shows numerous melt layers (Ahn et al., 2004;
Ahn & Brook, 2014; Das & Alley, 2005). Visible melt layers have been avoided during sampling for gas reconstructions for
both ice cores. According to Ahn et al. (2004), it can further not be excluded that the observed absolute offsets between $CO_2$
measurements from Siple Dome relative to EPICA Dome C may stem from in-situ production.

Though melt layers are frequently observed all over Greenland (Alley & Anandakrishnan, 1995; Chappellaz et al., 1997;
NEEM community members, 2013; Neftel et al., 1983), ice cores from central Greenland have been used for gas
reconstructions (Blunier & Brook, 2001; Schilt, Baumgartner, Schwander, et al., 2010). In higher-altitude ice cores, the
occurrence of melt features is likely limited to the Holocene and Last Interglacial warm periods, and records covering the Last
Glacial are affected to a lesser extent. However, $CO_2$ reconstructions from Greenland are considered unreliable due to



significant in-situ production of non-atmospheric $CO_2$, which takes place in the ice sheet over time even in the absence of liquid water (Tschumi & Stauffer, 2000).

## 4 Current applications of melt layers as proxies

The primary application of melt layers in ice cores is as qualitative and quantitative temperature proxy (Alley & Anandakrishnan, 1995; Kameda et al., 1995; Okamoto et al., 2011). It is based on the non-linear relationship between frequency and thickness of melt features and temperatures above freezing and has led to melt-temperature records from alpine to polar sites (Abram et al., 2013; Das & Alley, 2008; Fujita et al., 2019). Comparisons between melt layer and $\delta^{18}O$ records (Grumet et al., 2001; Henderson et al., 2006; Koerner et al., 1973), nearby weather stations and reanalysis data (Fujita et al., 2019) have been used to show the coherence of derived temperature records. Thereby, it has been noted that the occurrence of melt features depicts complex dynamics at any ice-core site (Alley & Anandakrishnan, 1995). While melt is assumed to reflect summer temperature conditions rather than annual mean temperature (Herron et al., 1981; Koerner, 1977a; Koerner & Fisher, 1990), the relevance of winter melt events is not to be underestimated for alpine to coastal polar sites (Zheng et al., 2020), especially since winter temperatures are rising more quickly than summer temperatures at mid-latitude and Arctic sites (Graham et al., 2017). An increasing number of melt layers can be caused (a) by an absolute temperature rise during one or several seasons of the year, (b) enhanced variability of temperatures during at least one season, or (c) a combination of both (Das & Alley, 2008). Changes of external parameters like elevation of ice cap surfaces can also affect the local climate and melt conditions (Das & Alley, 2008; Fig. 2). Furthermore, the near- and sub-surface firn conditions affect the prominence of melt features, with melt layers appearing more distinctly in cold firn than in homogeneously wetted, temperate firn (see Sect. 2.2. & 3.1.). All factors discussed in Sect. 2 affect the frequency and thickness of melt features and are helpful context for quantitatively interpreting melt layer profiles. Against this background, certain sites and glacier facies like the lower accumulation zone, where melting occurs but doesn't percolate through the entire annual layer, are more sensitive to the melt threshold and therefore more suitable sites for melt records than others (Abram et al., 2013). Given the uncertainty induced by percolation, Westhoff et al. (2022) have cautioned against the interpretation of melt layer profiles as annually resolved temperature proxy records and recommended ≥10-year averages. Using the dependency of melt occurrence on warm temperatures, Okamoto et al. (2011) have also been able to show that melt feature percentage correlates with terminus retreat rate of a neighbouring glacier.

Besides the simple occurrence of melt layers, significant potential lies in the structural characteristics of melt features as proxy for environmental conditions. Based on the physics of melt layer formation, structural characteristics of melt features are expected to differ dependent of meteorological setting and local climate, which leave an imprint on the near-surface snow. Early studies have shown that warm summers were characterized by coarser-grained superimposed ice than colder summers with fine-grained firn (Koerner, 1970; Koerner et al., 1973). Future studies of melt-affected ice cores could describe the



structural variability of melt features in ice cores in greater detail and thereby provide new insights into the processes shaping melt features.

Another emerging research area comes from the relationship between forest fires, consequential BC dust distribution and enhanced glacier melt (Kaspari et al., 2015; Keegan et al., 2014). Given that BC is a reliable tracer of forest fires even in melt-affected areas (Osmont et al., 2020), comparing BC and melt feature records could reveal new aspects for interpreting melt features as proxies.

## 5 Conclusions and outlook

Global temperature is rising and melting will affect a growing number of alpine to polar ice-core drilling locations, more frequently, with larger intensity, and at higher elevation further inland. For this reason, we have conducted a literature review, bringing together snow physics, firn hydrology and ice-core proxy research, for studying melt-affected ice cores.

Melt events are driven by local to global meteorology and climate, glacier surface characteristics like the albedo, and hydro-thermal conditions of the upper snow column. Melting and refreezing is governed by the surplus of energy available and follows approximately the 0°C isotherm. Identification of melt layers usually takes place either manually during ice-core processing, discriminating between melt-affected and unaffected firn sections, or using digital line-scanning, which provides higher-resolution insights into melt characteristics. Approaches like this show the diverse structural imprints of snowmelt on the ice-air matrix, ranging from homogeneously wetted horizons to distinct melt layers and preferential flow fingers. Those develop depending on the dynamic interplay of atmosphere and subsurface conditions and hold potential as environmental proxy.

Preferential elution is the primary issue of melt-affected ice-core chemistry. Dominated by solubility of the impurities in water and ice, a multitude of factors, including pre-melt snow chemistry, metamorphism and meltwater flow conditions, determines the availability of impurities to relocation or removal during melt events. Thus, the preservation of chemical species strongly differs with the study site. Where significant amounts of liquid water are present, the fractionation slope for stable water isotopes ($\delta^2$H/ $\delta^{18}$O) is reduced and affected records often show an amplitude reduction of the seasonal cycle, biased values and smoothing of the signal. Altered total air content and gas concentrations must be considered for melt-affected gas records in ice cores.

Despite these limitations of melt-affected ice cores, melt layers have been successfully applied as qualitative (above 0°C) and quantitative temperature proxy (using MFP as an indicator of surface temperature). Approaches ranging from using ion ratios (e.g. $Mg^{2+}/Na^+$) and the dissolved fraction percentage in the upper ice column to noble gases Xenon and Krypton in deep ice help to identify the extent of ice profiles affected by melting and refreezing.

Our literature study shows that melting poses both challenges and potential for ice-core interpretation and dating approaches, and also that the diversity of melt features and effects are of interest for comprehensive and site-specific investigation. Learning



from the advances of alpine ice-core science, the following adaptations have been shown to be helpful to ensure high-quality ice-core research under warming conditions.

1. (Re-)assessing the thermal and hydrological conditions of the subsurface prior to drilling helps to target ice-core analysis to predominantly cold, poly-thermal or temperate ice.

2. Sampling fresh (near-)surface snow throughout the seasonal cycle allows to establish a baseline of impurity levels
prior to melting, and in-situ snowmelt infiltration experiments using dye can be used to parametrize relocation depth site-specifically.

3. High-resolution structural analysis of ice cores (e.g. line scanning), more detailed melt logs including feature characteristics, and systematic assessments of melt indicators (e.g. DFP, $Cl^-/Na^+$ ion ratio) provide useful context to better understand chemical records.

4. Prioritization of impurities less mobile during melt events like ammonium, BC, insoluble and low-abundant (partially) water-soluble trace elements as dating tools and environmental proxies, and using multi-year averages where melt percolates beyond an annual layer leads to more reliable environmental reconstructions.

With this review we contribute to a growing body of literature aiming to foster a comprehensive understanding of signal
formation in ice cores and to improve their climatic interpretations. We hope to raise awareness among (polar) ice-core researchers about melt features in ice cores, how to identify and investigate them in greater detail, and how to gauge limitations and potential of melt-affected profiles for ice-core climate reconstructions.

*Author contributions.* DEM and ERT conceptualized the study. DEM organised the project and conducted the literature
analysis fundamental to this study. ERT and EW supervised the project. DEM prepared the initial draft, and all co-authors contributed to review and editing of the manuscript including figures.

*Competing interests.* The authors declare that they have no conflict of interest.

*Funding.* DEM was supported by BAS Cambridge, and the NERC C-CLEAR Doctoral Training Programme (grant no.NE/S007164/1). ERT received core funding from NERC to the British Antarctic Survey's Ice Dynamics and Palaeoclimate
programme. EW is supported by a Royal Society Professorship.

*Acknowledgments.* The authors want to thank Margit Schwikowski, who was consulted and provided experienced, external advice for Sect. 3.2. on melt-affected chemistry records. The melt layer images used in Fig. 1a and Fig. 1i have been reprinted from the Journal of Glaciology, and the melt layer image in Fig. 1f has been reprinted from the Annals of Glaciology with permission of the International Glaciological Society. The image used in Fig. 1d is reprinted from Global and Planetary Change
Volumes 84–85, Fisher et al., Recent melt rates of Canadian arctic ice caps are the highest in four millennia, 3–7 (2012), with permission from Elsevier. The melt layer sequence in Fig. 1h is reprinted from Nature Geoscience Volume 6, Abram et al.,



Acceleration of snow melt in an Antarctic Peninsula ice core during the twentieth century, 404–411 (2013), with permission from Springer Nature.

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
