# Peer review of "Review article: Melt-Affected Ice Cores for (Sub-)Polar Research in a Warming World"

_EGUsphere, 2023_

## Author Comment (AC1)

**Review 1**

General comments:

This is a great review paper on melt features in cores and should be published with some very minor changes. The authors do a great job in summarizing the importance of melt features and their effect on e.g. the climatic integrity of ice cores, which is of great importance to the ice core community.

I am missing a short explanation of the effect of melt in alpine and polar ice cores at the beginning of the manuscript. Similar to lines 577 to 581. This would be useful to distinguish when the authors refer to alpine or polar cores and to make it clear to the reader if the current section is valid for the full spectrum of melt events, i.e. massive melt events on coastal cores or tiny mm-melt events on the central Greenland ice sheet.

There are also some inconsistencies using the dash separating, e.g. "line scanning" and "line-scanning" throughout the manuscript.

For a better overview of the manuscript, I would suggest changing the list from (1)-(8) in lines 91 to 94 to the corresponding section numbers in the manuscript.

*Dear anonymous reviewer,*

*Thanks a lot for taking the time to critique this review article manuscript and for your positive feedback.*

*Your suggestion for the content outline at the end of the introduction is very much appreciated and has been re-emphasized by the second reviewer. We have changed the numbering in line 111-115, so that they match the section numbers: "We provide a detailed literature review regarding external drivers of melt events (Sect. 2.1); physics of melt layer formation and behaviour during snow metamorphism (Sect. 2.2); identification and quantification of melt (Sect. 2.3); structural characteristics of melt features (Sect. 3.1); effects of melting on records of chemical impurities, i.e. major ions, trace elements, black carbon (BC), and organic species (Sect. 3.2), stable water isotopic signatures (Sect. 3.3), and gas record (Sect. 3.4); applications of melt layers as environmental proxies (Sect. 3.5)."*

*A brief introduction to the varying effects of melt in alpine and polar ice cores has been added in revised line 49-52: "While melting is rare and related alterations of climate indicators are yet of little concern to some researchers working on ice cores from central Antarctica, key climate proxy records like $\delta^{18}O$ have already been obliterated through melting in numerous low-latitude high-altitude glaciers (Thompson et al., 2021) and are gradually deteriorating at (sub-)polar sites like Svalbard (Spolaor et al., 2023)."*

*Inconsistencies in spelling of line scanning have been streamlined to "line scanning" throughout the manuscript, including revised line 24 and 743. Line 288 remains the only exception because of line scan is used as attribute in "line-scan derived". In a similar way, the authors have re-read the manuscript for other spelling inconsistencies and corrected them where necessary.*

Specific comments:

Figure 1: I would prefer to see the scale explanation at the beginning of the caption.

*Thanks for this comment. We follow your suggestion to clearly explain the scale and orientation of each panel right from the start, and we have amended the figure caption as follows: "Figure 1: Examples of melt layers reported in snow and ice profiles from around the world using various*

*techniques; the grey scale bar at the bottom right of each panel equals 1 cm, and the top of each snow/firn/ice section is at the top of each panel: a) two melt layers, 4–6 mm thick, …"*

Figure 2: the map of Antarctica (1) is misleading for the figure, as the largest part of Antarctica is the plateau, which is not the main message of the figure.

*In the originally submitted version of figure 2, Antarctica is included as an example of a polar region, in which the climate and melt formation conditions are (among other factors) influenced by teleconnections and synoptic-scale atmospheric circulation as described in (revised) line 141-142 & 155-156. While the authors agree that central Antarctica is not the main area of melting, the intended message of this part of the figure is to symbolize the importance of large-scale meteorological patterns for melt formation, e.g. in coastal Antarctica. We therefore have added arrows to this map (see illustration below), which symbolize large-scale circulation in a simple way without going into the details provided in the text body.*

[Figure]

Line 132: "4 mm of melt" please specify if this is related to snow, firn, ice, or water equivalent.

*Thank you for this question. The estimate is in mm w.eq. and taken from Van den Broeke (2005), where they state: "A maximum melt rate of 4 mm per day occurs for flow from the NW, which represents a combination of warm air advection and a föhn effect caused by the mountains of the AP." For clarification, we have added "w.eq." to the estimate in line 140 of the revised manuscript.*

*For further explanation, which is beyond the scope of the manuscript text body, their estimate is methodically based on automatic weather station (AWS) temperature data from the Larsen Ice AWS, which Van den Broeke (2005) used to calculated melt hours (taking the temporal distribution of melt hours over the day and season into account). In combination with a simple expression for the magnitude of the energy fluxes at the atmosphere-ice shelf interface, from which they derive a meltwater flux per hour (0.53 mm per hour), they calculate the maximum estimate of 4 mm w.eq. cited above.*

Line 136-139: "Atmospheric rivers, especially during winter". Neff 2018 describes atmospheric rivers during summer and Wille et al 2019 both summer and winter. Please rephrase the sentence "especially during winter".

*From your comment, it seems like there may have been a misunderstanding about the statement made in former line 136-139. The authors absolutely agree that atmospheric rivers are not necessarily seasonally biased in occurrence. However, melt production appears strongly correlated with the occurrence of atmospheric rivers, and in Antarctica, this is especially the case during winter.*

*For example, Wille et al. (2019) state: "Atmospheric rivers are associated with around 40% of the total summer meltwater generated across the Ross Ice Shelf to nearly 100% in the higher elevation Marie Byrd Land and 40–80% of the total winter meltwater generated on the Wilkins, Bach, George IV and Larsen B and C ice shelves." Going into greater detail beyond the scope of our review article here, they state: "During these winter months (March–October), ARs and their residual moisture accounted for*

*40–60% of the melt days on the Larsen B and Larsen C ice shelves along the central portion of the eastern AP and represent 20–40% of the melt magnitude, according to MAR (Supplementary Figs. 2b and 3c). Further south along the outlet glaciers on the Wilkins Coast, 90–100% of winter melt days and melt magnitude are linked with AR activity."*

*Focussing on Greenland, Neff (2018) also "revealed that from 2000 to 2012, atmospheric rivers played an increasing role in driving summertime (June, July, August) melt and accumulation. This connection was particularly evident over the western GIS, where increased accumulation (snowfall) at higher elevations was unable to balance mass loss at the lower elevations through melting."*

*To emphasize the common result of Neff (2018) and Wille et al. (2019) that melt production is often associated with atmospheric rivers, we have rephrased the sentences in line 144-147 as follows: "Linear-shaped air intrusions are often referred to as 'atmospheric rivers' and correlate highly with melt production in both Greenland and Antarctica (Neff, 2018; Wille et al., 2019). The association is especially strong in Antarctica during winter (Wille et al., 2019), and for example, an atmospheric river is considered responsible for the observed extensive melt in West Antarctica in 2016 (Wille et al., 2019)."*

Line 184: please specify if coastal or central Antarctica.

*Apologies for the typo, the article by Laska et al. (2016) cited in line 191-192 of the revised manuscript deals with Svalbard melt conditions. More specifically, the estimate is based on AWS data from Hansbreen at an elevation of 179 m above sea level, and the conditions described are expected to resemble those at some coastal low-elevation sites in Antarctica and the Sub-Antarctic. The sentence has been corrected as follows: "This duration threshold is reached within a few hours to less than a day, as has been recorded for Hansbreen in Svalbard by Laska et al. (2016)."*

Line 198 and 230: "Infiltration beyond the current year is a source of uncertainty" implies that most melt is in the current year's snow layer. Line 230 states that melt does not tend to form at the surface but 50-100 cm deep, i.e. below the current year's snow layer. The two sentences seem to contradict each other, please clarify.

*Thank you for pointing out the need for further clarification in this section. The sentences in revised line 204-208 now read: "Meltwater infiltration into deeper, older snow makes percolation a secondary process leading to mixed snow compositions and climatic signatures, e.g. of summer melt in winter snow (Moore et al., 2005). For similar reason, infiltration beyond the current annual layer is an issue when interpreting ice-core proxy and melt layer records, so that using multi-year averages of melt indices has been recommended previously (Graeter et al., 2018)."*

*We have also added context for the information "50-100 cm deep" in revised lines 241-243 for clarification: "They [dye experiments] also highlight where melt layers don't form directly at the surface but at greater depth. At Neumayer Station in Dronning Maud Land, meltwater has been documented to refreeze 50–100 cm below the snow surface during summer (Kaczmarska et al., 2006)."*

Line 315 and 316: here it is not entirely clear if you refer to the snowpack or polar firn for the introduction of this section. Also, do you not analyze melt in deep ice because it does not penetrate that deep, or do you not analyze the effects of these old melt events? Please clarify.

*We have changed the wording in the first sentences of Sect. 3.1 to clarify both the focus of this section (physical and structural imprint of melting where it happens) and that melt features will still be present and detectable (though not visually) in deeper, clathrated sections of polar ice cores. Melt features do not decay so the impact of time since formation (age) is not a critical factor. However, differences in*

*concentration may diffuse to surrounding layers over time and smaller melt events may not be detectable anymore due to thinning. Revised lines 330-334 now read:*

*"Melting near the snow surface leaves a physical imprint on the stratigraphy of the glacier. Here, we discuss the appearance of melt features in snow, firn and bubbly ice, where structural differences are visible (Fig. 1). As these melt features are buried under new layers of snow, they participate in the snow/firn densification process as well as ice thinning at greater depth. Melt features in deep ice, where high hydrostatic pressure forces air bubbles into clathrates and bubble-free melt sections are practically impossible to detect visually, are addressed in Sect. 2.3.1 and Sect. 3.4."*

Table 1: there is a dash and a word missing at the very bottom left of page 14.

*The dimensions of the cell at the bottom left of page 15 in Table 1 have been corrected so that its full content is visible: "Measurements of snow and/or meltwater"*

Figure 5: a, b, and c are missing in the figure.

*Thanks for pointing this out. The figure panels are now labelled (a), (b), and (c).*

Line 499: is this "pre-melting" term the same as used, e.g., in line 181, or should it mean before/prior to melting?

*Here in revised line 522, "pre-melting" doesn't refer to the specific process discussed in former line 181 (equals revised line 188), so that we amended the phrase to "prior to melting" as you suggested. To avoid misinterpretations, we also checked the entire manuscript for the term "pre-melting" and made the same corrections in revised lines 461 and 749.*

Line 527: does "preserved" refer to a spatial sense in the snowpack? Please specify.

*Yes. We mean that the vertical profile of ammonium and other ions is better preserved than that of $Ca^{2+}$, $SO_4^{2-}$. To clarify this spatial dimension of the proxy records, we have added the words "vertical concentration profiles" to the sentence in revised line 550-551: "Among major ions, vertical concentration profiles of $NH_4^+$, $F^-$ and $Cl^-$ are generally better preserved than those of $Ca^{2+}$, $SO_4^{2-}$ (Table 1)."*

Line 663: do we not know exactly how many? Why is the number smaller or equal to 5?

*The authors agree that a more precise number would be favourable. However, the estimate "≤5 melt layers" in revised line 690 stems from Etheridge et al. (1996), who state: "At most five melt layers, less than 1 cm thick, were identified in each of the DE08 cores and even fewer in DSS." An exact number can therefore not be given here.*

Line 668: "frequently" makes it sound like a quasi-annual phenomenon, which is not the case. Please change the wording.

*Following your suggestion, we have rephrased the sentence in revised line 695 to: "Though melt layers are less rare in Greenland…*

Line 670: "higher-altitude", higher than in central Greenland? Please specify.

*To clarify that we are referring to high-altitude ice cores from central Greenland here, we have added the word "such" to the sentence in revised line 697-699: "In such high-altitude ice cores, the occurrence of melt features is likely limited to the Holocene and Last Interglacial warm periods, and records covering the Last Glacial are affected to a lesser extent."*

Line 712 " with larger intensity, and at higher elevation further inland." Change to: "… with larger intensity, at higher elevation, and further inland".

*Thanks for this suggestion. We have changed the wording in revised line 737-739 accordingly: "Global temperature is rising and melting will affect a growing number of alpine to polar ice-core drilling locations, more frequently, at higher elevation, and further inland."*

---

## Author Comment (AC2)

**Review 2**

The preprint 2023-1939 by Moser et al fills a current gap in the literature on the physics and chemistry of ice in the perspective of retaining climatic and environmental information out from coring ice fields that temporarily are effect by melt. Standard textbooks such as Takeo Hondoh´s book "Physics of Ice Core record" from 2000 are usually focused in cold firn ice cores (melting phenomena are mentioned in Hondoh´s book, and maybe worth a reference in this manuscript!), and the development of firn modeling and analytical technologies have opened new lanes of possibilities in extracting interesting and important information out from periodically melting / warm firn ice fields. This preprint is a result of a large, brave and commendable effort to collect and present a benchmark of where this field of science is today. With the current warming and its accelerated effect, it is of interest to retrieve information from these ice fields before the information gets to washed out by percolating melt and rain water. This work can such play an important role as a standard piece for new and old students in this field.

General comments

Since this is a review article I will focus at the organization of this work. My first reading of the preprint made me somewhat confused of where the review was aiming at. The general organization: 1 Introduction; 2 From melting to melt-affected ice cores; 3 Manifestations of melt in ice-core records; 4 Current applications of melt layers as proxies; 5 Conclusions and outlook makes sense from 1 to 3. Section 4 hang a bit outside the general structure, and could be added into section 3, or even placed into 2.3.1. Maybe assess the general organization again. The illustrations are another topic that caught my eye. Potentially there could be hundreds of illustrations, since the review cites a large amount of work. From what I can see the figures are used to illustrate important principles and key information in the review. The illustrations are perhaps also used to break to long sections of text, into more color and different texture to please the readers eye. If so, pages 6 -12 are lacking illustrations, and such are a bit more dense and "heavy" to get through. Try to anchor the figures better in the text; to take the opportunity to explain the information in figures and expand the relevance of the figures in the section. With this, the illustrations will bring in more information and will be used as to bring in additional knowledge, apart from the aesthetical value.

*Dear Veijo Pohjola,*

*We appreciate the time and effort that you dedicated to providing feedback on our manuscript and are grateful for your valuable comments to our paper. We have incorporated most of the suggestions, and they are both highlighted within the manuscript and explained in a point-by-point response below.*

*Regarding the general organization of the paper, we agree with your assessment that Sect. 4 on current applications of melt layers as proxies received special emphasis in comparison to the second-order sections on melt structure, chemistry, isotopes and gases (Sect. 3.1.-3.4.) by making it a first-level chapter. Following your advice, we have placed former Sect. 4 at the end of Sect. 3. Now numbered Sect. 3.5. in the 'Manifestations of melt in ice-core records' chapter, it is more obvious that melt features can be considered as environmental proxy in a similar way as the ones discussed in Sect.3.1.-3.4.. At the same time, keeping this content at the end of Sect. 3 maintains our train of thought from melt formation to manifestations and interpretation, which we hope is intuitively chronological from a researching point of view and thereby eases the reading experience.*

*As you say, we focus on providing illustrations of important principles. Since melt effects on chemistry have historically been more under debate than the effects on stable water isotope (SWI), the authors have intentionally dedicated a few figures to clarifying the related aspects. The preservation issues and factors relevant for SWI as known to date are partly covered in Fig.7, so that have added a direct reference to it in line 585 of the revised manuscript. Figure 5 further shows an exemplary melt-affected yet seasonally resolved SWI record from the Swiss Grenzgletscher, which we reference in line 597 of the revised manuscript. Beyond these two figures, the authors see no need for a separate illustration.*

*To spread the illustrations more evenly within the manuscript, we have moved some of the figures (Fig. 1, Fig. 2), within their respective section. To ensure closer ties between text body and illustrations, we have further added references to Fig.1 in line 274 and line 336 of the revised manuscript. In line 728, we now refer to Fig. 3 to re-emphasize the diversity of melt imprints. In addition, Figure 3 has been repositioned to revised line 386, so that the direct link between the sentence above and the figure is clearer. As the figure caption contains significant level of detail and is well-embedded in the reading flow, we think that a repetition of this content in the text body is not necessary. A reference to Fig. 4 has been added in line 427 of the revised manuscript. Another reference to Fig. 6 has been added in line 524 of the revised manuscript, and aside from that, it is already explained in detail in line 449-454 of the revised manuscript. While the paragraphs in lines 461-504 are already aligned with the content and structure of Fig. 7 (right 1-3), we have added more detailed references to the right part of the illustration in the respective sub-sections of the text body: (1) in line 470, (2) in line 476 and 490, (3) in line 497.*

Further comments:

1. How is the structure are lined up in the abstract as compared in the introduction, and further connected to the different sections. First, should the seven points in the abstract represent the different sections, and such display a "table of content"? Second, should these items of content be similar in the abstract and in the rows 90-95 in the introduction?

*Thank you for pointing out potential confusion about the numbering in abstract, introduction and manuscript sections. For clarification of the content given at the end of the introduction, we have aligned the numbering in line 111-115 with the section numbering. It now reads: "This paper provides a detailed literature review regarding external drivers of melt events (Sect. 2.1); physics of melt layer formation and behaviour during snow metamorphism (Sect. 2.2); identification and quantification of melt (Sect. 2.3); structural characteristics of melt features (Sect. 3.1); effects of melting on records of chemical impurities, i.e. major ions, trace elements, black carbon (BC), and organic species (Sect. 3.2), stable water isotopic signatures (Sect. 3.3), and gas record (Sect. 3.4); applications of melt layers as environmental proxies (Sect. 3.5).*

*The content structure in the abstract (line 15-18) is generally linked to the section structure of the paper in order to show the main outline, but it does not strictly copy the sequence of section titles to avoid losing track of the main topics by listing all sub-chapters and section numbers in the abstract. We keep this more detailed information for the introduction of the paper itself and have removed the numbers in the abstract to prevent confusion. Revised line 15-19 now read: "This review first covers melt layer formation, identification and quantification of melt, and structural characteristics of melt features. Subsequently, it discusses effects of melting on records of chemical impurities, i.e. major ions, trace elements, black carbon, and organic species, as well as stable water isotopic signatures, gas record, and applications of melt layers as environmental proxies."*

2. Could you consider to walk the reader through the arguments for the chosen organization of the manuscript after your presentation of aims and objectives?

*To explain the train of thought underlying the paper while maintaining brevity of this introduction, we have added the following clause in line 115-117: "Here, we review formation, manifestation, and interpretation of melt chronologically and focus on those aspects of near-surface melting, which are important for ice-core research."*

3. The ice chemistry section could be separated between particulate and ionized content. The analytic methods and the information out of these parameters are quite different, and they also bring different information, and behave differently during wet events.

*Thank you for this comment. Unfortunately, it is not always the case that different methods are used for particulate or ionic species, so that the existing literature only allows for an integrated discussion of the elution behaviour of various chemical species. In the case of trace elements analysed with ICP-MS, for example, everything from fully soluble (ionic) species to insoluble (particulate) species is measured. On the other hand, ion chromatography only assesses the ionized chemical content. While a more comprehensive analysis and discussion of elution behaviour of individual species in their various states is of interest, this requires a lot more research beyond the current state of literature and therefore is beyond the scope of this paper. Nevertheless, we agree that general differences in elution behaviour of particulate and water-soluble chemical impurities need to be mentioned, and our paper explicitly addresses them in Fig. 6 and the corresponding text body (revised line 444-455). Towards the end of Sect. 3.2 (line 550-566), we also lay out how various chemical species, which differ in characteristics and contained environmental information, can be used in the light of differing elution behaviour.*

4. The section of water isotopes is fairly short in comparison to the chemistry section. You could argue that the basic information is available in a number of texts, and refer to more basic literature of this sort, perhaps begin with the pioneering work by Dansgaard.

*Indeed, extensive publications on the general potential of stable water isotopes for ice core climate reconstruction have been published elsewhere, so that we only discuss alterations of SWI through melting and refreezing here. We refrain from adding many introductory literature references, which are not crucial for understanding here, but have amended the start of the SWI section (Sect. 3.3.) to keep the broader picture in mind. Revised line 574-578 now reads: "Isotopic fractionation is fundamental to the interpretation of stable water isotopes (SWI, Gat, 1996) as climate proxies in ice cores, and an extensive body of ice-core scientific literature starting with the pioneering work by Dansgaard (1964) exists on this topic. Here, we solely discuss alterations of SWI through melting and refreezing, because the fractionation of heavier and lighter isotopes takes place during phase changes, so that melting and refreezing in the snowpack plays a significant role (Koerner et al., 1973)"*

Minor comments

Li 12-14. Rephrase the sentence "Since coastal…..in alpine settings".

*From your recommendation to rephrase the sentence, we understand that our use of the terms "coastal low-elevation", "(sub-)polar" and "alpine" has been confusing here. This second sentence of the abstract is intended to briefly highlight that melt effects are an increasingly important issue for sub-/polar researchers, who are probably the main readership of this paper (revised line 15-16). In addition, it is important to the authors to indirectly acknowledge the longer-standing expertise of alpine ice core researchers in working with melt-affected ice core records from warmer conditions. In*

*the hope that this explanation helps, we have changed the wording in line 12-14 slightly: "Since (sub-)polar ice caps are crucial environmental archives but highly sensitive to ongoing climate warming, the Arctic and Antarctic research community is increasingly faced with melt-affected ice cores, which are already common in alpine settings of the lower latitudes."*

Li 14. Does not this review also include warming firn outside alpine settings?

*You are right. This review addresses the fundamental issues associated with melt-affected ice cores in a range of conditions. We mention alpine settings here, as alpine ice core research has had and found ways to deal with melt-affected ice cores, so that researchers who are less familiar with this topic, can learn from this. Furthermore, we expect sub-/polar researchers to be a key part of the readership of this paper, so that this part of the abstract points out the increasing relevance of melting in these regions. To cover both these aspects, the sentence in revised line 12-14 has been amended to: "Since (sub-)polar ice caps are crucial environmental archives but highly sensitive to ongoing climate warming, the Arctic and Antarctic research community is increasingly faced with melt-affected ice cores, which are already common in alpine settings of the lower latitudes."*

Li 97. On this row there is a short aim and objective, that can be expanded as commented in major comments above, if we, has the meaning authors, that line is badly posed, since I would presume it is the reader who wants to learn…

*Following your recommendation, we have amended the wording in line 111 of the revised manuscript to: "This paper provides a detailed review … ". In a similar way, we have changed revised line 118 of the revised manuscript to: "Finally, this study aims to learn from alpine ice-core scientists and contribute…"*

Li 128-131. May the sentence of temperature increase related to heat content be too trivial in The Cryosphere?

*We agree with your assessment and have shortened the paragraph to one amended sentence in line 138-139 of the revised manuscript: "Positive air temperatures are the primary, non-linearly related trigger of melt and a well-established, integrated proxy for melt intensity (Abram et al., 2013; Hock, 2003; Bell et al., 2018). "*

Li 133-149. This section on heat from advected air masses over ice fields is a bit exhaustive, and takes focus off the ice properties. Consider to shorten/cut this part.

*Thank you for this comment. We understand that this elaboration about meteorological factors for melt formation can seem less relevant in comparison to the structural and chemical assessment of melt-affected ice cores as presented in the following chapters. However, we see it as fundamental context for interpreting ice core records, which can be melt-affected by a variety of conditions. Given the close link of meteorology and the physics of melting (Sect.2.2.), cutting this part could take away from the basis of ice-core climate reconstruction. For this reason, we have removed one longer sentence in revised line 152 and otherwise kept the main elements of this section.*

Li 184. Typo? "Therefor"

*Thank you. The spelling in line 192 of the revised manuscript has been corrected to: "Therefore, modelling of melt…"*

Li 635-639. I would guess that gas content and gas evacuation capacity is dependent on how the ice crystal boundaries are configured in ice matrix, and how the effective porosity varies in the firn/ice matrix.

*The reviewer assumes correctly that the Total Air Content (TAC) depends, among other factors, on the pore space volume of the firn in the lock-in zone. If a melt feature is present in a given volume of firn in the lock-in zone, it will have less open pore space because of the volume occupied by solid ice instead of porous firn. This therefore leads to a lower TAC as described in revised lines 659-664 in the Gas Section. We have revised this section for improved clarity:*

*"Lastly, total air content (TAC) is often interpreted as driven by, among other effects, the elevation of the glacier surface, i.e. higher elevation is associated with lower TAC as the barometric air pressure decreases with altitude (Delmotte et al., 1999; Martinerie et al., 1992). However, a reduced TAC may also be indirectly caused by a melt feature because the presence of impermeable ice, i.e. the refrozen melt layer, has replaced porous firn and thereby lowered the volume of pore space in melt-affected firn in the lock-in zone. This may lead up to 40% reduction in TAC (Stauffer et al., 1985), even though this reduction does not seem to be consistent throughout different melt features and ice cores."*

*Note that the ice crystal boundary configuration beyond this point does not impact the TAC. This is because all available TAC measurement techniques extract 100% of the trapped air from the ice sample, independent of the ice crystal structure.*

---

## Author Response (AR2)

Dear Kerim Nisancioglu,

Thanks a lot for your editorial support with this manuscript.
We have made two minor technical corrections:

- Title and abstract:
  As the main focus of this paper is on using knowledge from ice cores from lower latitudes to tackle the increasing challenges of a warming climate for analysis of polar ice cores, and the examples are mostly from polar sites (Greenland, Antarctic, and Svalbard), we have decided to remove *"(sub-)"* from the title and abstract (line 12 and line 15) for clarity.

- Caption of Figure 1:
  To clearly state the obtained rights to reproduce the digital and line scan images used in Fig. 1d and 1h, we have added the phrase "*reproduced with permission from Elsevier/SNCSC*" in the caption of Fig.1 (line 73 and 78).